# Antimicrobial and Immunomodulatory Properties and Applications of Marine-Derived Proteins and Peptides

**DOI:** 10.3390/md17060350

**Published:** 2019-06-12

**Authors:** Hee Kyoung Kang, Hyung Ho Lee, Chang Ho Seo, Yoonkyung Park

**Affiliations:** 1Department of Biomedical Science, Chosun University, Gwangju 501-759, Korea; hkkang129@gmail.com; 2Department of Convergences, Kongju National University, Kongju 314-701, Korea; cinenote27@naver.com (H.H.L.); chseo@kongju.ac.kr (C.H.S.); 3Research Center for Proteineous Materials, Chosun University, Gwangju 501-759, Korea

**Keywords:** marine organism, immunomodulatory peptide, protein hydrolysates, immune response

## Abstract

Marine organisms provide an abundant source of potential medicines. Many of the marine-derived biomaterials have been shown to act as different mechanisms in immune responses, and in each case they can significantly control the immune system to produce effective reactions. Marine-derived proteins, peptides, and protein hydrolysates exhibit various physiologic functions, such as antimicrobial, anticancer, antioxidant, antihypertensive, and anti-inflammatory activities. Recently, the immunomodulatory properties of several antimicrobial peptides have been demonstrated. Some of these peptides directly kill bacteria and exhibit a variety of immunomodulatory activities that improve the host innate immune response and effectively eliminate infection. The properties of immunomodulatory proteins and peptides correlate with their amino acid composition, sequence, and length. Proteins and peptides with immunomodulatory properties have been tested in vitro and in vivo, and some of them have undergone different clinical and preclinical trials. This review provides a comprehensive overview of marine immunomodulatory proteins, peptides, and protein hydrolysates as well as their production, mechanisms of action, and applications in human therapy.

## 1. Introduction

The immune response plays a crucial role in maintaining human health by identifying and killing pathogens, aging cells, or tumor cells. Its functions can be affected by many factors, including pathogen presence, tissue injury, and cardiac infarction [1]. Immunomodulation refers to the ability of the immune system to control various life-threatening diseases, such as cancer, human immunodeficiency virus, multiple sclerosis, and aging [2,3,4]. Macrophages are mononuclear leukocytes, which function as primary defenders of the host and can recognize and eliminate microbial pathogens and cancer cells via phagocytosis. These cells are also involved in immune regulation by producing cytokines, such as interferon (IFN)-γ, tumor necrosis factor (TNF)-α, interleukin (IL)-1β, IL-6, IL-10, and IL-12. Clinically, immunomodulators can be classified as immunoadjuvants, immunostimulants, and immunosuppressants. Immunotherapy is the treatment of disease by modulating the host’s immune system. Currently, numerous drugs are used clinically to control human immune function, including levamisole, imiquimod, pidotimod, tilorone, cyclophosphamide, prostaglandin, cyclosporine A, thiocarbamate, niridazole, and penicillamine [5,6,7,8,9,10,11,12,13,14]. However, most synthesized immunomodulatory drugs exhibit toxicity and side effects, limiting their use to cases such as those of chronic diseases [15]. In contrast, the majority of naturally-derived immunomodulatory proteins or peptides do not show side effects and are less costly, suggesting their potential for use in immunotherapy.

The marine environment is rich in bioactive resources, but few studies have examined the ability of these resources to modulate the immune response. Approximately 2,210,000 species are thought to exist in the ocean, but only about 190,000 species have been recorded [16,17]. Peptides, proteins, and protein hydrolysates isolated from marine algae and fish can function as immunostimulants. The potential immunomodulatory activity of marine derivatives and their scientific analyses are now emerging. This review describes the immunomodulatory role of marine-derived biomaterials.

## 2. Brief Overview of the Immune System

The immune system is a host defense system that serves to remove potentially harmful substances known as pathogens. The immune system is divided into two subsystems: the innate and adaptive immune systems. 

Innate immunity (natural or native immunity) is non-specific and provides the first line of defense through the skin, mucosal tissue, bone marrow (neutrophils, eosinophils, basophils, mast cells, monocytes, dendritic cells, and macrophages), or inflammatory components (cytokines, interferons, and defensins). Macrophages and neutrophils are important first-line responders in phagocytosis [18].

Adaptive immunity (specific or acquired immunity) is highly specific to dangerous pathogens and is the body’s second line of defense involving T lymphocytes and B lymphocytes. The adaptive immune response can be either humoral or cell-mediated. Humoral immunity is mediated by B lymphocytes, which release antibodies specific for the infectious pathogen. The cell-mediated response involves binding of T lymphocytes to the pathogen or infected cells, followed by lysis of these cells and the secretion of immune regulatory factors, such as cytokines. The three subtype of T lymphocytes are cytotoxic T cells, helper T (TH) cells, and regulatory T (suppressor T) cells. Cytotoxic T cells express a surface receptor, cluster of differentiation (CD)8+, which recognize endogenous antigens associated with major histocompatibility complex class I and kill infected cells. TH cells express the surface receptor CD4+ and recognize exogenous antigens complexed with major histocompatibility complex class II. TH cells secrete cytokines and help activate B and T lymphocytes and other immune cells [19]. Regulatory T cells help to control the immune response by preventing harmful immune activation and maintain tolerance, or prevent autoimmune diseases by maintaining self-tolerance.

## 3. Immunomodulatory Compounds from Marine Organisms

Over the past few decades, more than 16,000 marine organisms have been isolated from the oceans and have been widely studied. Marine-derived substances, such as proteins, peptides, glycoproteins, polysaccharides, and lipids show potential for controlling the immune system [20,21,22]. This review summarizes the developmental status of marine-derived immunomodulatory proteins, peptides, and protein hydrolysates from different organisms.

### 3.1. Immunomodulatory Proteins and Amino Acid

Marine biomaterials (proteins, enzymes, oligosaccharides, biopolymers, fatty acids, minerals, and pigments, etc.) described come from a variety of marine natural sources. Marine biomaterials contain large amounts of diverse proteins (10–47% (*w*/*w*)) with various bioactivities and functions. Many proteins in hemolymphs and hemocytes play important roles in the innate immune system by preserving several immune components, such as metalloproteins, glycoprotein, amino sulfonic acid, antimicrobial peptides (AMPs), protease inhibitors, and coagulation factors. Table 1 summarizes the immunomodulatory proteins and amino sulfonic acid (taurine) found in diverse marine biomaterials.

#### 3.1.1. Hemocyanins

Hemocyanins are metalloproteins that use copper-binding sites to bind and transport oxygen in a variety of mollusks. Hemocyanins, like hemoglobin, are multi-subunit molecules with a functional subunit that binds oxygen. Hemocyanins isolated from several marine gastropods show potent immunostimulatory effects against certain cancers with tolerable side effects observed in a murine colon carcinoma model [23,24]. Hemocyanins have been used as innate immunostimulators to suppress the development of superficial bladder cancer, while hemocyanins obtained from *Concholepas concholepas* have been evaluated in clinical trials for treating superficial bladder cancer [25]. Keyhole limpet hemocyanin is an immunostimulator derived from circulating glycoproteins of the marine mollusk *Megathura crenulata*. Keyhole limpet hemocyanin interacts with T cell monocytes, macrophages, and polymorphonuclear lymphocytes to improve the host immune response (Figure 1) [26,27]. Hemocyanin isolated from a gastropod *Fissurella latimarginata* is a novel immunostimulator with higher immunostimulatory activities than traditional hemocyanins, such as keyhole limpet hemocyanin and *Concholepas concholepas*, which were isolated from a gastropod [28].

#### 3.1.2. Lectins

Lectin, a glycoprotein that controls immune function, has been widely reported through innate immune responses leading to pathogen recognition and endocytosis, and adaptive immune responses such as B and T activation, and apoptosis [29]. Lectins are found in many different plants and animals. Lectins isolated from clam are homologous and modulate the innate immune response [30]. Three types of lectins have been identified: Ca^2+^-dependent (C-type), R-type, and metal independent galectins. C-type lectins recognize broad endogenous ligands and pathogens that regulate intercellular interactions and play a major role in carbohydrate recognition during the immune response [31]. These proteins are also very important in autoimmune diseases. Tachylectin, isolated from hemocytes of the horseshoe crab *Carcinoscorpius rotundicauda*, recognizes pathogens [32]. Microbes recognition occurs via phagocytosis or the lectin pathway in the complement system. The lectin from the mussel *Mytilus trossulus* stimulates the induction of proinflammatory cytokines such as TNF-α and IFN-γ but reduces the anaphylaxis of the anti-inflammatory cytokine IL-10 in human peripheral blood cells (Figure 2) [33]. Lectins, derived from the hemolymph of crustaceans, are considered as active precursors of antibodies because they play an important role in antimicrobial activity as well as immune recognition [34].

#### 3.1.3. Taurine

Taurine (2-amino ethane sulfonic acid) is an amino acid that is widely distributed in animal tissues including in the marine clam (Figure 3). Marine organisms are a rich source of taurine, but it is also produced by many other organisms. Taurine has cytoprotective and immunomodulatory effects and is enriched in immune cells including lymphocytes, monocytes, and neutrophils [35]. This suggests that high taurine levels in phagocytes mediating inflammatory lesions indicate a role in innate immunity [36]. Taurine accumulates in phagocytes; after contacting a pathogen, activated cells (neutral and macrophages) produce toxic oxidants and various antibacterial substances through the peroxidase system, and act at inflammatory sites to kill pathogens. Toxic oxidants are components of the innate immune system and protect the host from infections by killing pathogens, but they may also damage the host tissue. Taurine also acts as a scavenger to remove unwanted or harmful substances from the cells and protect cells from oxidative stress. Taurine modulates the immune system by activating NF-κB, a potent signal transducer for inflammatory cytokines [35]. Additionally, peroxisome proliferator-activated receptor-γ (PPAR-γ) is activated by taurine in the liver to control the regulation of glucose metabolism and adiposeness. Thus, the activation of PPAR-γ by taurine protects retinal neuronal damage in diabetic retinopathy [37].

### 3.2. Antimicrobial and Immunomodulatory Peptides

Studies of functional peptides that control immune responses are actively being conducted [38]. Although immunomodulatory peptides are known to enhance immune responses such as lymphocyte proliferation, natural killer (NK) cell activity, and cytokine regulation, the specific mechanisms of these activities remain unclear [39]. For example, the interactions of immunomodulatory peptides with the human nervous, digestive, cardiovascular, and immune systems are not well-understood [40]. For this reason, it is necessary to understand the interaction between peptides and these systems for the development of specific immunomodulatory peptides.

Marine-derived AMPs are defense molecules with the potential to enhance innate immunity [41]. The discovery that marine AMPs target specific cells confirmed the potential of these molecules as drug candidates [42]. Therefore, marine-derived AMPs are important sources of molecules with immunological regulatory properties. Table 2 and Table 3 summarize immunomodulatory peptides that have been identified in diverse marine sources.

#### 3.2.1. Callinectin

Callinectin is a proline- and arginine-rich AMP composed of 32 amino acids, including four cysteine residues, and shows antibacterial activity against gram-negative bacteria. Callinectin was isolated from hemocytes of the blue crab *Callinectes sapidus and* Mediterranean mussel *Mytilus galloprovincialis* with the amino acid sequence WNSNRRFRVGRPPVVGRPGCVCFRAPCPCSNY-NH_2_ (Figure 4) [43]. Callinectin has three isoforms, hydroxy-*N*-formylkynurenine group, *N*-formylkynurenine, and hydroxyl tryptophan, and has diverse functional groups according to changes in the tryptophan residues. Tryptophan modification of callinectin isoforms were reported in the AMPs of marine animals [44]. Callinectin strongly binds to anti-callinectin-like peptide antibodies in blue crab hemocytes and shows the potential for immunomodulation [43].

#### 3.2.2. Clavanin A and Clavanin-MO

Clavanin A (VFQFLGKIIHHVGNFVHGFSHVF-NH_2_) was isolated from hemocytes of the marine tunicate *Styela clava* [45]. Clavanin A shows broad antimicrobial activity both in vitro and in vivo against Gram-negative and Gram-positive drug-resistant bacteria and fungi. Clavanin-MO (FLPIIVFQFLGKIIHHVGNFVHGFSHVF-NH_2_) was synthesized by adding five hydrophobic amino acids to the N-terminus of clavanin A, which improved its cell interaction and ability to penetrate cell membranes compared to clavanin A (Figure 5). Clavanin-MO shows antibacterial activity against *Escherichia coli* and *Staphylococcus aureus*, and has been shown to suppress inflammatory responses that cause sepsis and destroy certain biofilms [46]. Both clavanin A and clavanin-MO affect components of the immune system and influence the inflammatory response through their immunomodulatory properties in C57BL/6 mice. The peptides increased the level of IL-10, an anti-inflammatory cytokine, and decreased the levels of IL-12 and TNF-α, two pro-inflammatory cytokines that boost inflammation and may lead to excessive damage [46].

#### 3.2.3. Crustin

Crustins, found in crustaceans, are cysteine-rich AMPs with a typical whey acidic protein (WAP) domain and play an important role in innate immune mechanisms [47]. The WAP domain is a conserved motif containing twelve cysteine residues with six disulfide bonds (Figure 6). Crustins with specific activity against marine Gram-positive bacteria *Corynebacterium glutamicum* have been reported in various crustaceans such as *C. maenas*, *Pacifastacus leniusculus*, *Scylla paramamosain*, and *Penaeus monodon* (MW 7–14 kDa). Crustins are released from the hemocytes of crustaceans by exocytosis [47,48,49,50,51].

#### 3.2.4. Defensin

Defensins are small cysteine-rich cationic AMPs that act as host defense peptides (Figure 7). Defensin was found in various sources, including animals, plants, and insects. The human defensins are classified into the α-defensins and β-defensins on the basis of their sequence homology and their cysteine residues. The first marine defensin was isolated by acidified gill extraction from *Crassostrea virginica* [52,53]. Defensins are antimicrobial peptides that disrupt the membrane of microbial pathogens and play a major role in immunomodulation by acting in the innate and adaptive immune response [54]. Marine-derived defensins were isolated from the oysters *C. gigas* and *C. virginica* and mussels *Mytilus edulis* and *M. galloprovincialis*. Three types of defensins (Cg-Defs) were identified in the pacific oyster *C. gigas* (*Cg-defh1* and *Cg-defh2*) and hemocytes. Two defensins, MGD1 and MGD2, derived from *M. galloprovincialis* showed potent antimicrobial activity by activating the immune response [55]. Following bacterial infection, an increase of MGD-1 in *M. galloprovincialis* plasma and MGD-2 stimulated the release from hemocytes. 

#### 3.2.5. Myticin

Myticin is also a cysteine-rich AMP derived from the Mediterranean mussel *M. galloprovincialis*. It has three isoforms, myticin A, myticin B, and myticin C. Myticin A (4.438 Da) and B (4.562 Da) were identified in the hemocytes and plasma of *M. galloprovincialis* and showed antibacterial activity against Gram-positive bacteria [56]. Myticin C showed antibacterial activity against the fungus *Fusarium oxysporum* and *E. coli* as well as Gram-positive bacteria and acted as an immunomodulator in vivo [57]. RT-qPCR analysis revealed immune-related Myticin C gene expression following in vivo immunostimulation in mussels, indicating its important role in innate immune defense (Figure 8) [58].

#### 3.2.6. Mytilin

Mytilins are cysteine-rich cationic AMPs found in marine mollusks. Mytilin A and mytilin B, isoforms mytilin C, mytilin D, and mytilin G1 were isolated from *Mytilus edulis* and *M. galloprovincialis* (Figure 9). All mytilin isoforms show potent antimicrobial activities [59,60]. Mytilins are transported through hemocytes to reach bacteria, and cells containing mytilin act as phagocytosing bacteria to prevent microbes from entering the circulatory system [60].

#### 3.2.7. Mytimycin

Mytimycin is an antifungal peptide (6233.5 Da) isolated from the blue mussel *Mytilus edulis* and *M. galloprovincialis* [55,59]. Mytimycin, which consists of 12 cysteines connecting six disulfide bridges and an C-terminal extension that contains an EF-hand domain (Ca^2+^ binding motif), defends against invading pathogenic microbes. The gene responsible for mytimycin is mainly expressed in circulatory hemocytes (Figure 10) [61].

#### 3.2.8. Phosvitin-Derived Peptide (Pt5)

Zebrafish phosvitin-derived peptide Pt5, consisting of the C-terminal 55 residues of phosvitin, has been shown to have antimicrobial activity and immunomodulatory activity comparable to phosvitin in vitro. Amino acid sequence of Pt5 was SRMSKTATIIEPFRKFHKDRYLAHHSATKDTSSGSAAASFEQMQKQNRFLGNDIP (Figure 11). Pt5 has been reported to increase the survival rate of zebrafish infected by *Aeromonas hydrophila* by significantly decreasing the number of *A. hydrophila* in the blood, spleen, kidneys, liver, and muscles. Pt5 also inhibits the expression of proinflammatory cytokine genes (IL-1β, IL-6, TNF-α, and IFN-γ) in the spleen and head kidneys of *A. hydrophila*-infected zebrafish, but increased the expression of anti-inflammatory cytokine genes (IL-10 and IL-14) [62].

#### 3.2.9. *Salmo Salar* NK-Lysin-Derived Peptides

NK-lysin are AMPs composed of 74 to 78 residues and that contain six cysteine residues that form three disulfide bonds and a C-terminal region that contains a saposin B-type domain (Figure 12). Peptide activity depends on these intact disulfide bonds, and the antimicrobial activity of NK-lysin was inhibited when the peptide was pre-treated with dithiothreitol [63]. NK-lysin from Atlantic salmon (*Salmo salar*) induced the expression of proinflammatory cytokines (IL-1β and IL-8) in the *S. salar* head kidney leukocytes. NK-lysin modulates the immune response, suggesting its potential for enhancing the immune response in fish [64].

#### 3.2.10. Scygonadin

Scygonadin (10.8 kDa) is an anionic AMP isolated from the seminal plasma of the mud crab *Scylla serrata* (Figure 13) [65,66]. Scygonadin are AMPs involved in the host defense by protecting the reproductive system of organisms [66].

#### 3.2.11. Thalassospiramides A and D

Thalassospiramides A and D, which are cyclic lipopeptides, were isolated from *Thalassospira* sp. Thalassospiramide D differs in structure at the *N*-terminus in which the proteinogenic serine residue in thalassospiramide A is replaced with the nonstandard phenylalanine-based statine residue 4-amino-3-hydroxy-5-phenylpentanoic acid (Figure 14) [67]. Thalassospiramides A and D suppressed lipopolysaccharide (LPS)-induced nitric oxide (NO) production by murine macrophage RAW 264.7 cells [68]. Thalassospiramides A and D were inhibited by IL-5, which plays an important role in TH2-mediated inflammatory diseases such as asthma.

#### 3.2.12. Tilapia Piscidin 3 (TP3) and Tilapia Piscidin 4 (TP4)

The tilapia piscidins are a group of peptides with antimicrobial, wound-healing, and antitumor functions. TP3 (FIHHIIGGLFSVGKHIHSLIHGH) and TP4 (FIHHIIGGLFSAGKAIHRLIRRRRR) are AMPs isolated from *Oreochromis niloticus* (Figure 15). TP3 and TP4 significantly increased the expression of several immune-related genes in muscle (IL-1β, IL-6, IL-8 TGF-β, and IκB) and decreased the expression of Toll-like receptor 5 (TLR5) after *Vibrio vulnificus* infection. Infection with *Streptococcus agalactiae* significantly decreased IL-1β, IL-8, TLR5, TGF-β, and IκB. TP3 and TP4 show potential for development as drug candidates to combat fish bacterial infections in aquaculture [69].

### 3.3. Immunomodulatory Protein Hydrolysates

Protein hydrolysates derived from various proteins have been reported to have a wide range of biological activities, such as anti-inflammatory, anticancer, antioxidant, antimicrobial, anti-hypertensive, and immunomodulatory activities [17,20,70,71]. During protein hydrolysis, peptide bond cleavage results in the formation of bioactive peptides with different sizes. Several proteolytic enzymes were successfully used to produce immunomodulatory protein hydrolysates. These enzymes include pancreatin, Kojizyme^TM^, trypsin, Alcalase®, Flavourzyme®, Protamex^TM^, α-chymotrypsin, pepsin, Neutrase®, and thermolysin. 

Immunomodulatory protein hydrolysates have not been reported to be cytotoxic, unlike protein hydrolysates with antibacterial or anticancer activity [70]. Antimicrobial peptides are important in the first line of the host defense system against pathogenic microorganisms that easily come in contact with the host through the environment [52,53]. Antimicrobial peptides from marine protein hydrolysates are increasingly isolated and reported during the last few years [72,73,74,75,76]. Since there are few reports of marine-derived protein hydrolysates having both antibacterial and immunomodulatory activity, this review summarizes the immunomodulatory protein hydrolysates isolated from diverse marine sources (Table 4).

#### 3.3.1. Chlorella Protein Hydrolysate

An enzymatic protein hydrolysate from the green microalga *Chlorella vulgaris* was prepared by hydrolysis of an ethanol-extracted cell biomass with pancreatin (Sigma-Aldrich, St. Louis, MO, USA). Both innate and specific immune responses (such as bone marrow cellularity and leukocyte counts in peripheral blood) were recovered when *Chlorella* protein hydrolysate was orally administrated in undernourished BALB/c mice, including significant increases in the lymphocyte pool, production of T-cell dependent antibody reactions, and reconstruction of delayed-type hypersensitivity reactions [77]. Starved mice treated with *Chlorella* protein hydrolysate showed a larger number of peritoneal exudate cells and higher activation of macrophages compared to non-supplemented mice. Additionally, stimulation of the mononuclear phagocytic system occurred as carbon clearance increased in the peripheral blood.

#### 3.3.2. Ecklonia Protein Hydrolysate

An enzymatic protein hydrolysate from the brown seaweed Alariaceae *Ecklonia cava* was prepared by hydrolysis of cell biomass with Kojizyme (Novo Nordisk, Bagsvaerd, Denmark). *Ecklonia cava* hydrolysate was shown to activate or suppress immune cell functions of murine splenocytes *in vitro.* ICR mice injected with *Ecklonia* protein hydrolysate showed enhanced splenocyte proliferation and increased numbers of splenocytes, lymphocytes, monocytes, and granulocytes. The numbers of CD4+ T cells, CD8+ T cells, and CD45R/B220+ B cells also increased. Additionally, TNF-α and IFN-γ, which are type Th1 cytokines, were downregulated by *Ecklonia* protein hydrolysate, while the type TH2 cytokines IL-4 and IL-10 were upregulated. Thus, *Ecklonia* protein hydrolysate has immunomodulatory effects and activates anti-inflammatory responses [78].

#### 3.3.3. Porphyra Protein Hydrolysate 

An enzymatic protein hydrolysate from the algae *Porphyra columbina* was prepared by hydrolysis with Alcalase® (Danisco S.A., Arroyito, Cordoba, República Argentina), trypsin, and a combination of both proteases. The hydrolysate showed immunosuppressive effects in rat splenocytes by enhancing anti-inflammatory cytokine (IL-10) production, while the production of pro-inflammatory cytokines such as TNF-α and IFN-γ was decreased [79].

#### 3.3.4. *Porphyra columbina* Protein Hydrolysate

An enzymatic protein hydrolysate from *P. columbina* was prepared by hydrolysis with flavourzyme (Sigma-Aldrich, St. Louis, MO, USA) and fungal protease concentrate. *P. columbina* protein hydrolysate was enriched in aspartic acid, alanine, and glutamic acid. It showed immunomodulatory effects on primary splenocytes, macrophages, and T lymphocytes in vitro. IL-10 secretion was increased in splenocytes (235%), macrophages (150%), and lymphocytes (472%) following treatment with *P. columbina* protein hydrolysate, while the production of TNF-α, IL-1β, and IL-6 by macrophages was inhibited (15–75%). The effect of *P. columbina* protein hydrolysate on IL-10 occurred through JNK, p38 MAPK, and NF-κB pathways in T cells [80].

#### 3.3.5. Edible Red Algae Protein Hydrolysate

An enzymatic protein hydrolysate from the edible red algae *Porphyra tenera* was prepared by hydrolysis with four proteases (Alcalase®, Flavourzyme®, Neutrase®, and Protamex^TM^) and seven carbohydrases [amyloglucosidase (AMG), Celluclast®, Dextrozyme®, Maltogenase, Promozyme, Termamyl®, and Viscozyme®]. These enzymatic hydrolysates showed antioxidant, anti-acetylcholinestrase (AChE), and anti-inflammation activities. Edible red algae protein hydrolysate showed no cytotoxicity in RAW264.7 macrophages, and inhibited LPS-induced NO production in RAW264.7 macrophages [81]. Therefore, edible red algae protein hydrolysate shows potential as a source of anti-inflammatory drugs.

#### 3.3.6. Edible Microalgae *Spirulina* Protein Hydrolysate 

An enzymatic protein hydrolysate from the filamentous blue-green algae *Spirulina maxima* was prepared by hydrolysis with trypsin, pepsin, and α-chymotrypsin (Sigma-Aldrich, St. Louis, MO, USA). Two peptides, LDAVNR (P1, 686 Da) and MMLDF (P2, 655 Da), from edible microalgae *Spirulina* protein hydrolysate significantly inhibited RBL-2H3 mast-cell degranulation by decreasing histamine release and increasing intracellular Ca^2+^. The inhibitory activity of P1 by blocking Ca^2+^- and microtubule-dependent signaling pathways, and the suppression of P2 were involved in phospholipase Cγ activation and reactive oxygen species production. Additionally, the inhibitory effect of P1 and P2 on the generation of IL-4 occurred via decreased nuclear factor-κB translocation [82].

#### 3.3.7. Oyster Peptide-Based Enteral Nutrition Formula 

An enzymatic peptide-based enteral nutrition formula from the oyster *Crassostrea hongkongensis* was prepared by hydrolysis with bromelain, pepsin, and trypsin. The immunological effects of an oyster peptide-based enteral nutrition formula using malabsorption mice and cyclophosphamide-induced immunosuppression mice were investigated, and spleen lymphocyte proliferation and NK cell activity was observed to be enhanced. This indicates that an oyster peptide-based enteral nutrition formula has an immune-stimulating effect on mice [83].

#### 3.3.8. Oyster Protein Hydrolysate

An oligopeptide-enriched protein hydrolysate oyster *C. gigas* was prepared by hydrolysis with protease from *Bacillus* sp. SM98011. The growth of implantable sarcoma-S180 was inhibited in a dose-dependent manner in BALB/c mice injected with oyster protein hydrolysate. Additionally, the body weight of BALB/c mice was further reduced by oral administration of oyster protein hydrolysate, while the weight coefficients of the thymus and spleen, activity of NK cells, spleen proliferation of lymphocytes, and growth of macrophages in mice with S180 increased markedly after administration of oyster protein hydrolysate. These results showed that oyster protein hydrolysate has strong immunostimulation activity in mice, which may lead to antitumor activity [15].

#### 3.3.9. Paphia Undulata Meat Protein Hydrolysate

An enzymatic protein hydrolysate from the Chinese clam *Paphia undulata* was prepared by hydrolysis with alkaline protease from *Bacillus subtilis*. The isolated fractions (P2 and P3) of *P. undulata* meat protein hydrolysate contained the peptides PHTC, VGYT, EF, LF, EGAL, WI, or WL, respectively. Amino acid analysis of P2 and P3 confirmed that the DPPH radical scanning activity was strong because of the high levels of hydrophobic amino acids, including leucine, phenylalanine, valine, and tryptophan. In addition, *P. undulata* meat protein hydrolysate enhances spleen lymphocyte proliferation activity [84].

#### 3.3.10. *Cyclina sinensis* Protein Hydrolysate (Novel Pentadecapeptide)

An enzymatic protein hydrolysate from the bivalve mollusk *Cyclina sinensis* was prepared by hydrolysis with pepsin. *Cyclina sinensis* protein hydrolysate with a molecular weight of less than 3 kDa showed immunomodulatory activity with the highest relative proliferation rate in RAW264.7 macrophages. The amino acid sequence of this novel pentadecapeptide is RVAPEEHVEGRYLV (1750.81 Da) and its immunomodulatory activity was found to result in enhanced macrophage phagocytosis, increased productions of NO, TNF-α, IL-6, and IL-1β, and up-regulation of the levels of iNOS, NF-κB, and NLRP3 in RAW264.7 cells (Figure 16). Thus, this protein hydrolysate shows potential for immunomodulation because it can facilitate macrophage activity by activating the NF-κB signaling pathway [85].

#### 3.3.11. *Ruditapes* Protein Hydrolysate

An enzymatic protein hydrolysate from short-necked clam, *Ruditapes philippinarum*, was prepared by hydrolysis with eight proteases (Alcalase®, Flavourzyme®, Neutrase®, Protamex^TM^, α-chymotrypsin, papain, pepsin, and trypsin). These enzymatic hydrolysates showed NO-inhibitory activity. Among the purified peptides in the *Ruditapes* protein hydrolysate, the NO-inhibitory peptide consisted of 10 amino acid residues (QCQAVASAV, 876 Da) at the N-terminal region (Figure 17). Additionally, purified peptides inhibited NO production in LPS-stimulated RAW264.7 cells. Purified peptides in the *Ruditapes* protein hydrolysate showed strong anti-inflammatory activity [86].

#### 3.3.12. Shellfish *Mytilus* Protein Hydrolysate

An enzymatic protein hydrolysate from shellfish, *Mytilus coruscus*, was prepared by hydrolysis with eight proteases (Alcalase®, Flavourzyme®, Neutrase®, Protamex^TM^, α-chymotrypsin papain, pepsin, and trypsin). Among the purified peptides in the *Mytilus* protein hydrolysate, the anti-inflammatory peptide consisted of 10 amino acid residues (GVSLLEEFFL, 1151.37 Da) at the N-terminal region (Figure 18). Additionally, purified peptides were found to inhibit NO production in LPS-stimulated RAW264.7 cells. Purified peptides in the shellfish *Mytilus* protein hydrolysate show anti-inflammatory activities [87].

#### 3.3.13. Alaska Pollock Protein Hydrolysate

An enzymatic protein hydrolysate from Alaska pollock (*Theragra chalcogramma*) was prepared by hydrolysis with trypsin (Sigma-Aldrich, St. Louis, MO, USA). The molecular weight of this hydrolysate is 622 Da and amino acid sequence is PTGADY. The hydrolysate was confirmed to have immunomodulatory activity, resulting in increased production of IL-2, IL-4, and IL-6 in immunosuppressed mice. Additionally, purified Alaska pollock protein hydrolysate significantly enhanced humoral, cellular, and non-specific immunity in immunosuppressed mice [88].

#### 3.3.14. Alaska Pollock Frame Protein Hydrolysate

An enzymatic protein hydrolysate from Alaska pollock (*T. chalcogramma*) were prepared by hydrolysis with seven proteases (alkaline protease, bromelain, Flavourzyme®, mixed enzymes for animal proteolysis, neutral protease, papain, and trypsin). The amino acid sequences of Alaska pollock frame protein hydrolysates are NGMTY (584 Da), LGLAP (470 Da), and WY (305 Da). These protein hydrolysates show immunomodulatory activity with the highest relative lymphocyte proliferation activity. The main amino acid residues in the purified Alaska pollock frame protein hydrolysate were proline, aspartate, glutamic acid, and leucine, and the peptides enriched with the main amino acid residues affected immunomodulatory activity [89]. 

#### 3.3.15. Fermented Pacific Whiting Protein

The fish protein concentrate from pacific whiting *Merluccius merluccins* were prepared by fermentation followed by proteolysis. In mice injected with fermented pacific whiting protein, the number of immunoglobulin A (IgA) cells increased in the small intestine laminaria propria, but not in the bronchial tissues. Significant increases in IL-4, IL-6, and IL-10 were observed in the laminar propria of mice injected with fermented pacific whiting protein. Pro-inflammatory cytokines (IFN and TNF-α) also increased, while intestinal homeostasis was maintained and no tissue damage was observed [90].

#### 3.3.16. Chum Salmon Oligopeptide Preparation

An enzymatic protein hydrolysate from the chum salmon *Oncorhynchus keta* was prepared by hydrolysis with complex protease. The molecular weight distribution of chum salmon oligopeptide preparations was 300–860 Da, and its main amino acid composition was glutamic acid, aspartic acid, lysine, leucine, arginine, and glycine. Chum salmon oligopeptide preparation greatly improved lymphocyte proliferation induced by mitogen concanavalin A, the number of plaque-forming cells, NK cell activity, the ratio of CD4+ TH cells in the spleen, and the secretion of TH1 (IL-2, IFN-γ) and TH2 (IL-5, IL-6) cytokines in female ICR mice. No differences were observed in weight gain, lymphoid organ indices, and phagocytosis capacity. Chum salmon oligopeptide preparation was confirmed to enhance the immune response of the host [91].

#### 3.3.17. Salmon Fish Protein Hydrolysate

Protein hydrolysate from Atlantic salmon was prepared by hydrolysis with endogenous hydrolyzing agents. The hydrolysates contained 60–70% di/tri peptides (less than 10 kDa). Following ingestion of salmon fish protein hydrolysate to malnourished children with grade I and II (Gomez’s classification), the immunoglobulin, CD4/CD8 ratio, and hemoglobin levels measured as immunological parameters did not vary significantly. Thus, salmon fish protein hydrolysate may be useful as a safe nutrient supplement for malnourished children [92].

#### 3.3.18. Salmon Byproduct Protein Hydrolysate

Enzymatic protein hydrolysate from salmon byproduct protein from the pectoral fin was prepared by hydrolysis with six proteases (Alcalase®, Flavourzyme®, Neutrase®, Protamex^TM^, pepsin, and trypsin). Salmon byproduct protein hydrolysate showed potent DPPH and hydrogen peroxide scavenging activities in a dose-dependent manner. Among the protein hydrolysates, salmon byproduct protein hydrolysate 1, which showed the highest antioxidant action among purified salmon byproduct protein hydrolysates, had a molecular weight of 1000–2000 Da. Its antioxidant amino acids (such as tyrosine, phenylalanine, proline, alanine, histidine, and leucine) account for 28.62% of the total amino acid content. Salmon byproduct protein hydrolysate 1 shows no cytotoxicity in Chang liver or RAW264.7 macrophage cells and inhibits intracellular reactive oxygen species generation, lipid peroxidation, and glutathione levels in in Chang liver cells. Salmon byproduct protein hydrolysate 1 also shows anti-inflammatory activity by inhibiting NO production and proinflammatory cytokines (TNF-α, IL-6, and IL-1β) in RAW264.7 cells [93].

#### 3.3.19. Salmon Pectoral Fin Byproduct Protein Hydrolysate

Enzymatic protein hydrolysates from salmon byproduct protein from the pectoral fin were prepared by hydrolysis with pepsin. All salmon pectoral fin byproduct protein hydrolysates showed anti-inflammatory activities. Among the protein hydrolysates, the tripeptide (PAY), which had a molecular weight of 349.15 Da, exhibited strong NO- and prostaglandin E2 (PGE2)-inhibition activity in LPS-stimulated RAW264.7 macrophages. Additionally, PAY significantly inhibited the protein expression of inducible NO synthase and cyclooxygenase-2 responsible for generating NO and PGE2. PAY treatment also inhibited the production of inflammatory cytokines (TNF-α, IL-1β, and IL-6) [94].

#### 3.3.20. Shark-Derived Protein Hydrolysate

Enzymatic protein hydrolysates from shark-derived protein (PeptiBal^TM^, innoVactiv, Inc., Rimouski, Canada) were prepared by hydrolysis with trypsin and α-chymotrypsin. All shark-derived protein hydrolysates had molecular weights below 10 kDa. Following oral administration of shark-derived protein hydrolysate, intestinal barrier function was enhanced by increasing the production of IgA and intestinal cytokines (IL-6 and TNF-α). Increased TGF-β and IL-10 contributed to the uncontrolled inflammatory response caused by infection with enterotoxigenic *E. coli* H10407. This confirmed that shark-derived protein hydrolysate can be used as a pharmaceutical agent to reduce the risk of bacterial infections and inflammatory-related diseases [95].

#### 3.3.21. Sweetfish-Derived Protein Hydrolysate

Enzymatic protein hydrolysates from sweetfish-derived proteins were prepared by hydrolysis with pepsin, trypsin, and α-chymotrypsin. Sweetfish-derived protein hydrolysate inhibited the production of NO, inflammatory cytokines (TNF-α and IL-6), and PGE2 in LPS-stimulated RAW264.7 cells. Moreover, sweetfish-derived protein hydrolysate inhibited the mRNA expression levels of inflammation-mediated proteins and inhibition of NF-κB activation. These results suggest that sweetfish-derived protein hydrolysates can be used as anti-inflammatory agents [96].

#### 3.3.22. Common Carp Egg Protein Hydrolysate

Enzymatic protein hydrolysates from common carp (*Cyprinus carpio*) egg were prepared by hydrolysis with Alcalase®, pepsin, and trypsin. The molecular weights of the three sweetfish-derived protein hydrolysates were 5–90 kDa and contained high levels of essential amino acids and docosahexaenoic acid among ω-3 fatty acids. The three sweetfish-derived protein hydrolysates increased the proliferation of spleen lymphocytes, serum IgA, spleen NK cytotoxicity, and mucosal immunity (secretory IgA), as well as induced spleen CD4+ and CD8+ cells in female BALB/c mice. The results confirmed that sweetfish-derived protein hydrolysates can improve immune system function [97].

#### 3.3.23. Rohu Egg Protein Hydrolysate

Enzymatic protein hydrolysates from rohu (*Labeo rohita*) egg were prepared by hydrolysis with Alcalase®, pepsin, and trypsin. The molecular weights of rohu egg protein hydrolysates were less than 10 kDa. Rohu egg protein hydrolysate increased splenic NK cell cytotoxicity, macrophage phagocytosis, serum IgA levels, mucosal immunity (secretory IgA) in the gut, and percentages of spleen CD4+ and CD8+ cells in BALB/c mice. The results confirmed that rohu egg protein hydrolysates can improve immune function [98].

## 4. Marine Immunomodulatory Peptide-Based Drug Therapeutics and Future Prospects 

Immunomodulatory protein, peptides, or protein hydrolysates act on a variety of targets, containing monocytes, macrophages, NK cells, T and B lymphocytes, CD4+ and CD8+ T cells, and CD45R/B220+ B cells (Table 1, Table 2 and Table 3). The mechanisms of action of marine-derived immunomodulatory proteins, peptides, or protein hydrolysates mainly affects by macrophages activation; phagocytosis stimulation; increased number of leukocytes; increased production of NO, immunoglobulins, and cytokines; splenocyte proliferation; NK cell stimulation; and activation of the NF-κB- and MAPK-dependent pathways (Figure 19).

The number of elderly people in the population is increasing. As people age, immune system function decreases and inflammation increases. This leads to a greater number of infections and higher death rates [99]. Innate immune responses are affected by various factors. During aging, cytokine production by monocytes and macrophages is altered, phagocytotic capacity is decreased, and TLR expression is reduced [100]. Marine-derived proteins, peptides, and protein hydrolysates were found to be capable of controlling immune functions [90]. Thus, marine-derived proteins, peptides, and protein hydrolysates palliated weakening of the immune system, thus improving the quality of life and reducing medical costs as well as contributing to immunity.

## 5. Conclusions

Bioactive proteins, peptides, or protein hydrolysates from marine biomaterials, particularly their immunomodulatory effects, have been widely studied. A recent increase in the incidence of various diseases has prompted attempts to use various immunomodulatory agents to control these diseases. Most studies have focused on evaluating immune effects of marine-derived biomaterials in cellular culture or animal studies, and subsequent studies in humans are rare. Therefore, additional clinical studies are needed to investigate the safety, biocompatibility, and immune effects of immunomodulatory products containing proteins, peptides, and hydrolyzed proteins developed from marine-derived biomaterials in the human body. The effects of immunomodulation by marine-derived proteins or peptides require detailed investigation.

## Figures and Tables

**Figure 1 marinedrugs-17-00350-f001:**
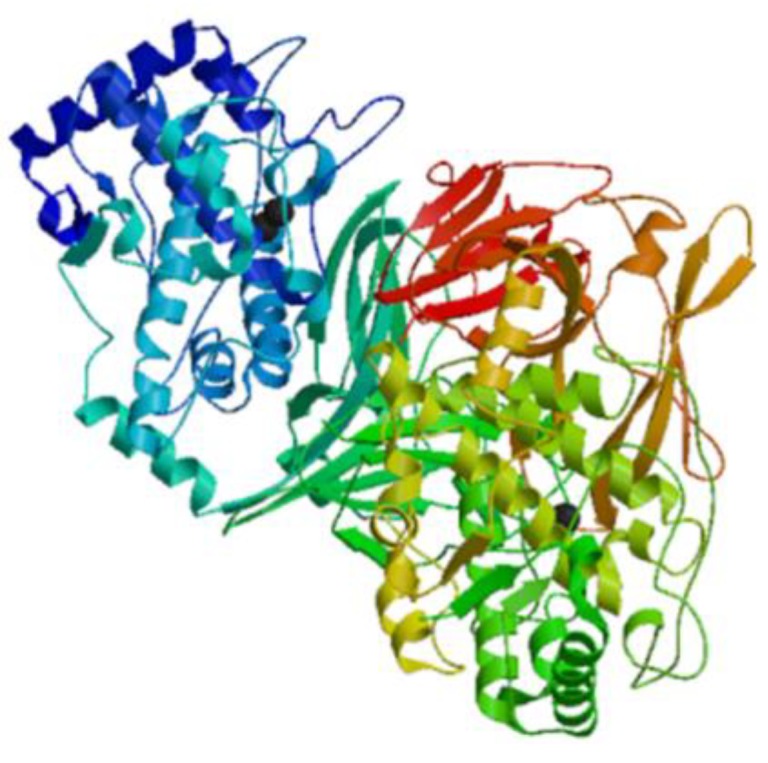
Structural model of keyhole limpet hemocyanin chain A from the marine mollusk *Megathura crenulata* (Genbank: 3L6W_A) created using the SWISS-MODEL server (Swiss Institute of Bioinformatics, Basel, Switzerland, https://swissmodel.expasy.org/) [26].

**Figure 2 marinedrugs-17-00350-f002:**
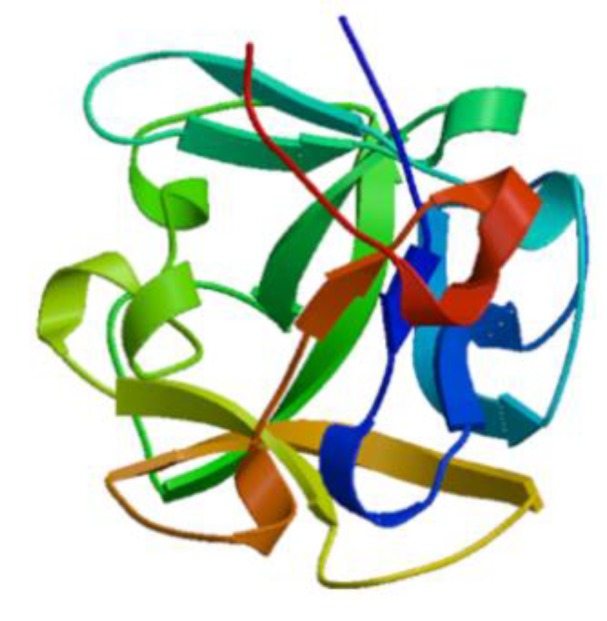
Structural model of lectin from the mussel *Mytilus trossulus* (Genbank: AKI29293.1) created using the SWISS-MODEL server (https://swissmodel.expasy.org/) [33].

**Figure 3 marinedrugs-17-00350-f003:**
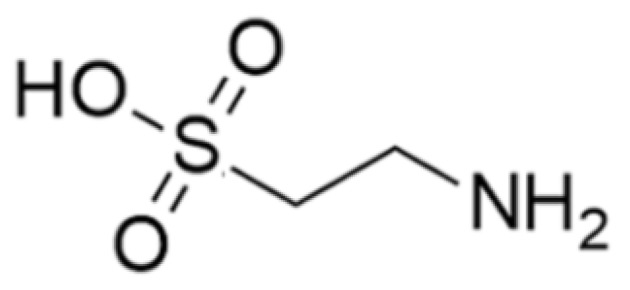
Molecular structure of taurine created using the Chemdraw (PerkinElmer, Waltham, MA USA) [35].

**Figure 4 marinedrugs-17-00350-f004:**
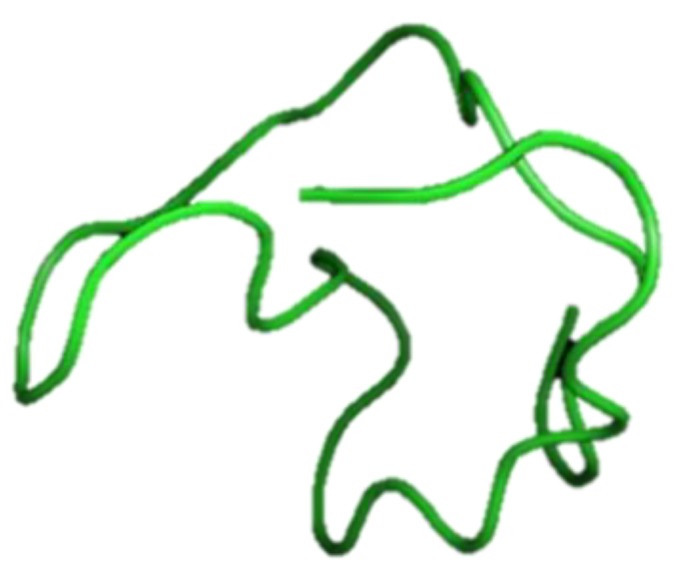
Structures of callinectin. Simulations of the three-dimensional structures were obtained from http://mobyle.rpbs.univ-paris-diderot.fr/cgi-bin/portal.py#forms::PEP-FOLD [43].

**Figure 5 marinedrugs-17-00350-f005:**
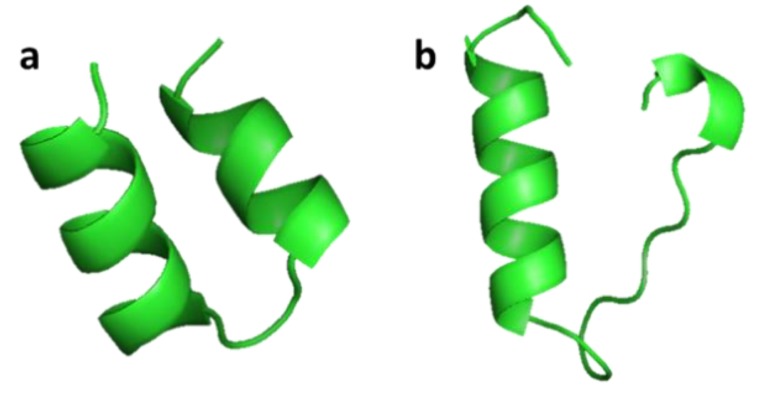
Structures of clavanin A (a) and clavanin-MO (b). Simulations of the three-dimensional structures for A and B were obtained from http://mobyle.rpbs.univ-paris-diderot.fr/cgi-bin/portal.py#forms::PEP-FOLD [46].

**Figure 6 marinedrugs-17-00350-f006:**
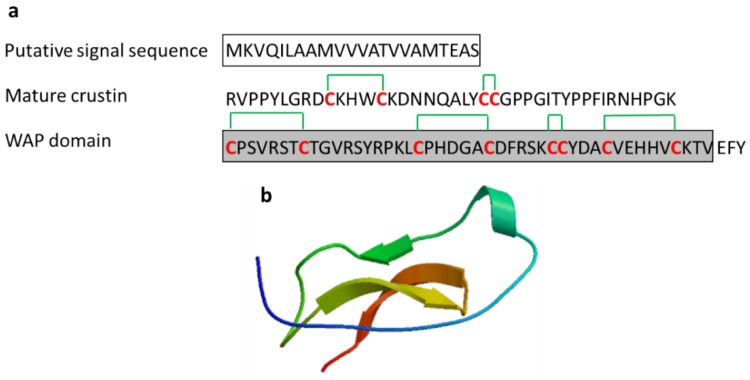
Amino acid sequences of crustin from the haemocyte of the mud crab *Scylla serrata* (Genbank: ADW11096.1) [47]. (**a**) Amino acid residues in the open square box indicate a putative signal sequence. Cysteine residues that participate in the formation of intramolecular disulfide bonds are red characters and the WAP domain is denoted in the gray box. Six disulfide bonds indicated green lines. (**b**) Structural model of active crustin of *S. serrata* created using the SWISS-MODEL server (https://swissmodel.expasy.org/).

**Figure 7 marinedrugs-17-00350-f007:**
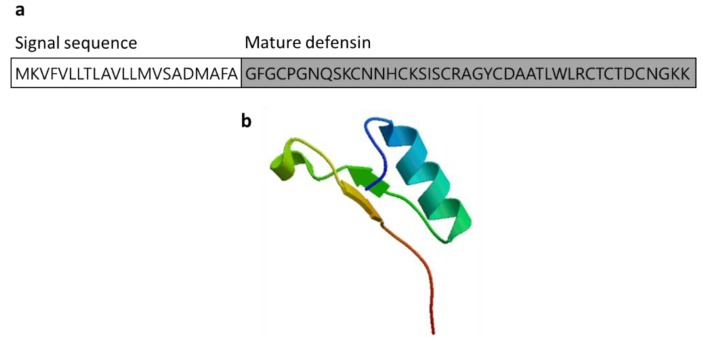
Amino acid sequences of defensin from the oysters *Crassostrea gigas* (Genbank: ACQ76262.1) [52]. (**a**) Amino acid residues in the open square box indicate a putative signal sequence. The active defensin is denoted in the gray box. (**b**) Structural model of defensin of *C. gigas* created using the SWISS-MODEL server (https://swissmodel.expasy.org/).

**Figure 8 marinedrugs-17-00350-f008:**
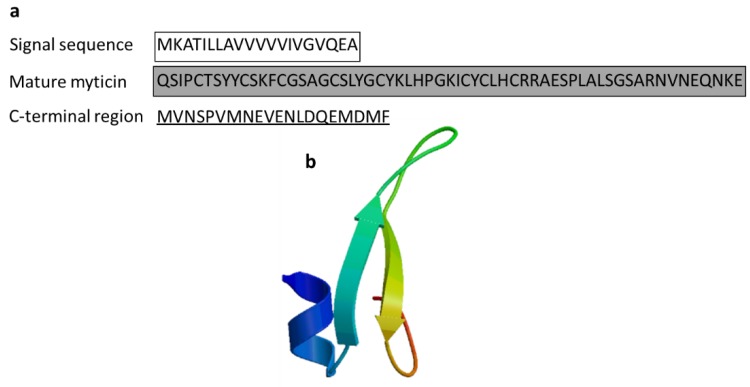
Amino acid sequences of myticin C from the Mediterranean mussel *Mytilus galloprovincialis* (Genbank: AEZ79080.1) [58]. (**a**) Amino acid residues in the open square box indicate a putative signal sequence. The mature peptides are denoted in the gray box. Black-lined amino acid residues indicate C-terminal regions. (**b**) Structural model of myticin C of *M. galloprovincialis* created using the SWISS-MODEL server (https://swissmodel.expasy.org/).

**Figure 9 marinedrugs-17-00350-f009:**
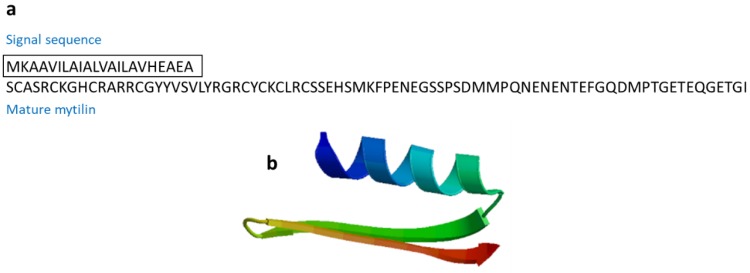
Amino acid sequences of mytilin B from the Mediterranean mussel *Mytilus galloprovincialis* (Genbank: AAD45013.1) [59]. (**a**) Amino acid residues in the open square box indicate a putative signal sequence. (**b**) Structural model of active mytilin B of *M. galloprovincialis* created using the SWISS-MODEL server (https://swissmodel.expasy.org/).

**Figure 10 marinedrugs-17-00350-f010:**
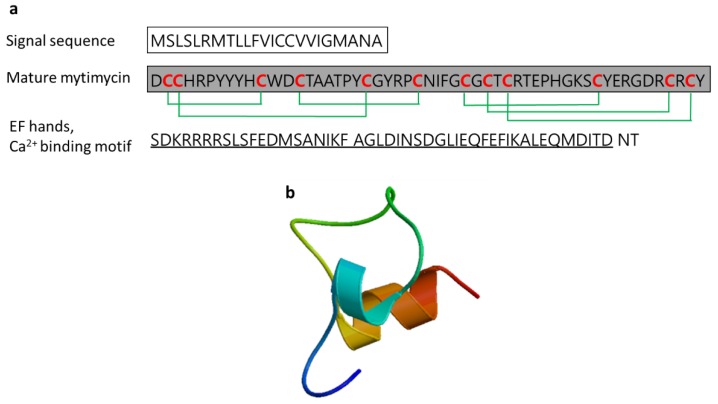
Amino acid sequences of mytimycin from the blue mussel *Mytilus edulis* (Genbank: AET85056.1) [61]. (**a**) Amino acid residues in the open square box indicate a putative signal sequence. Cysteine residues that participate in the formation of intramolecular disulfide bonds are red characters and the mature peptide is denoted in the gray box. Six disulfide bonds are indicated by green lines. Black-lined amino acid residues indicate EF hands, Ca^2+^ binding motif. (**b**) Structural model of active mytimycin of *M. edulis* created using the SWISS-MODEL server (https://swissmodel.expasy.org/).

**Figure 11 marinedrugs-17-00350-f011:**
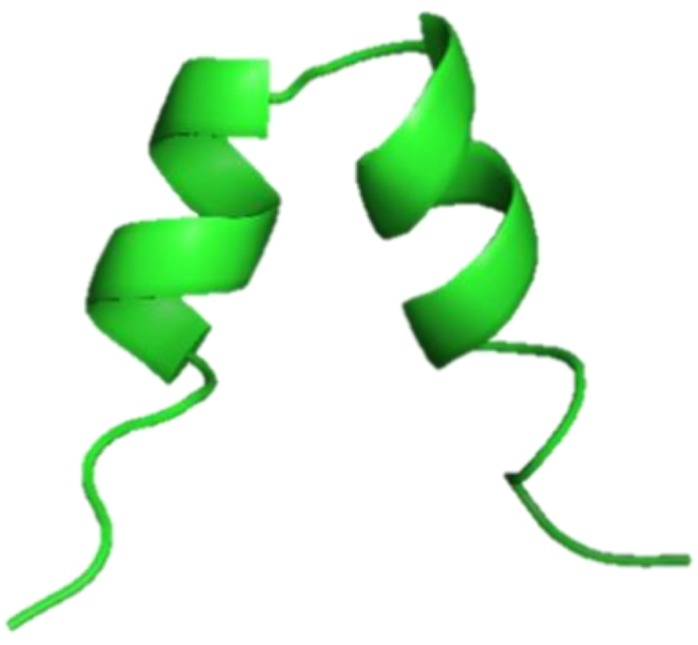
Structural model of Zebrafish phosvitin-derived peptide Pt5, created using the SWISS-MODEL server (https://swissmodel.expasy.org/) [62].

**Figure 12 marinedrugs-17-00350-f012:**
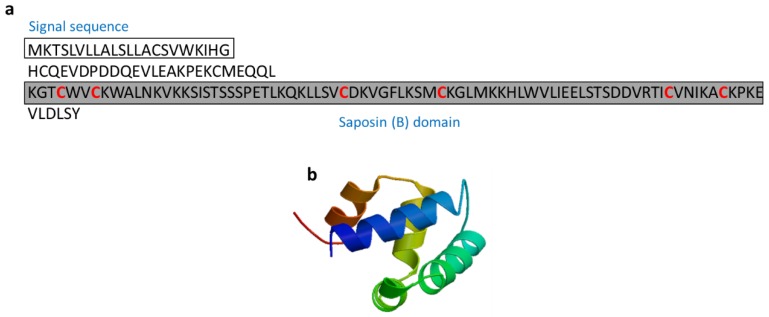
Amino acid sequences of NK-lysin from Atlantic salmon (*Salmo salar*) (Genbank: XP_013985382) [63]. (**a**) Amino acid residues in the open square box indicate a putative signal sequence. Cysteine residues that participate in the formation of intramolecular disulfide bonds are red characters and the saposin B domain is denoted in the gray box. (**b**) Structural model of active NK-lysin (saposin B) created using the SWISS-MODEL server (https://swissmodel.expasy.org/).

**Figure 13 marinedrugs-17-00350-f013:**
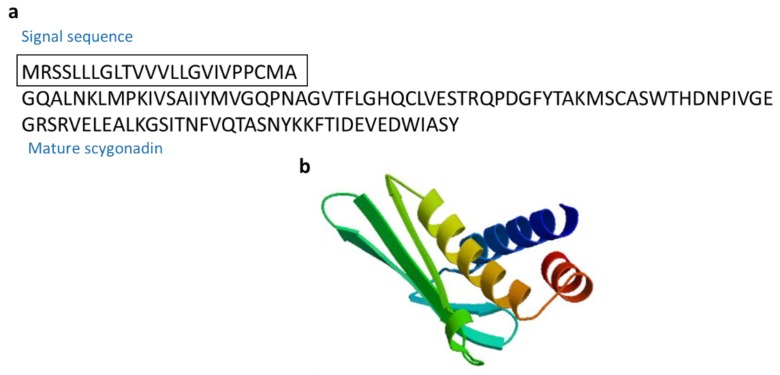
Amino acid sequences of scygonadin from the seminal plasma of the mud crab *Scylla serrata* (Genbank: AAW57403.1) [65]. (**a**) Amino acid residues in the open square box indicate a putative signal sequence. (**b**) Structural model of active scygonadin of *S. serrata* created using the SWISS-MODEL server (https://swissmodel.expasy.org/).

**Figure 14 marinedrugs-17-00350-f014:**
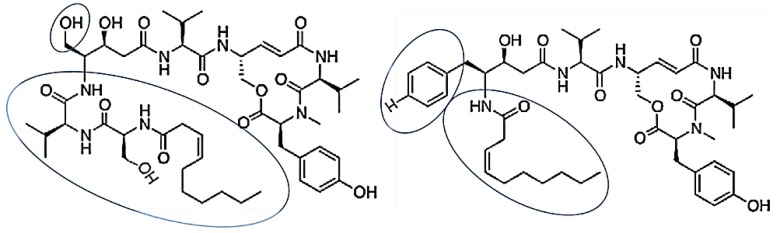
Chemical structures of thalassospiramide A (**a**) and thalassospiramide D (**b**) created using the Chemdraw. Lipopeptide side chain indicated by two elliptical circles [67,68].

**Figure 15 marinedrugs-17-00350-f015:**
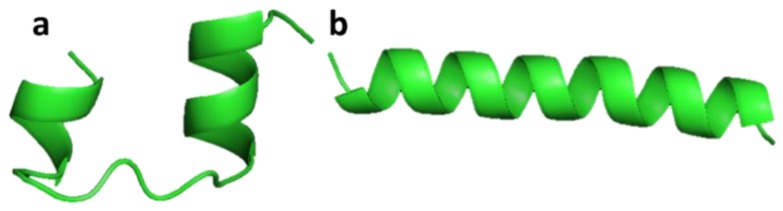
Structures of TP3 (**a**) and TP4 (**b**). Simulations of the three-dimensional structures for A and B were obtained from http://mobyle.rpbs.univ-paris-diderot.fr/cgi-bin/portal.py#forms::PEP-FOLD [69].

**Figure 16 marinedrugs-17-00350-f016:**
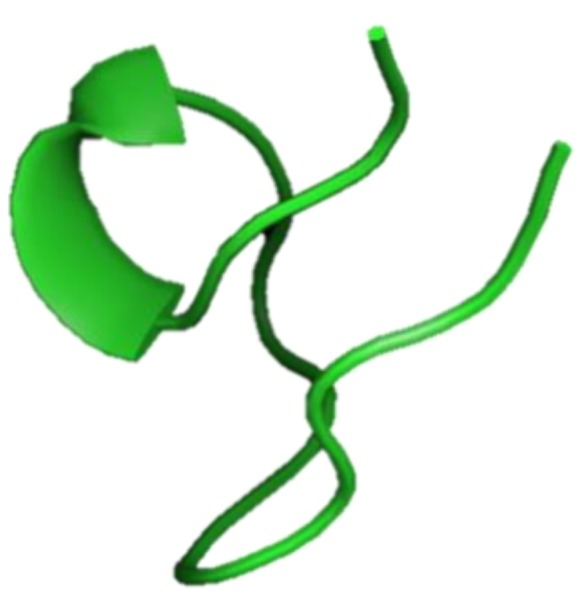
Structures of *Cyclina sinensis* protein hydrolysate (novel pentadecapeptide) from the bivalve mollusk *Cyclina sinensis*. Simulations of the three-dimensional structures were obtained from http://mobyle.rpbs.univ-paris-diderot.fr/cgi-bin/portal.py#forms::PEP-FOLD [85].

**Figure 17 marinedrugs-17-00350-f017:**
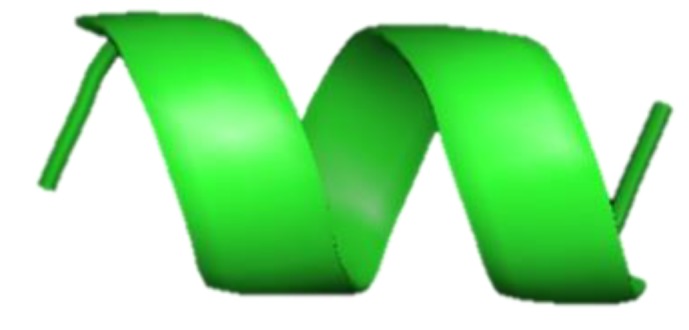
Structures of *Ruditapes* protein hydrolysate from short-necked clam, *Ruditapes philippinarum*. Simulations of the three-dimensional structures were obtained from http://mobyle.rpbs.univ-paris-diderot.fr/cgi-bin/portal.py#forms::PEP-FOLD [86].

**Figure 18 marinedrugs-17-00350-f018:**
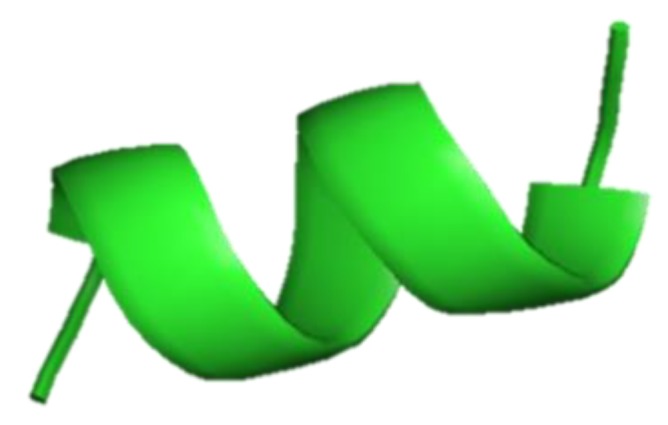
Structures of shellfish *Mytilus* protein hydrolysate from shellfish, *Mytilus coruscus*. Simulations of the three-dimensional structures were obtained from http://mobyle.rpbs.univ-paris-diderot.fr/cgi-bin/portal.py#forms::PEP-FOLD [87].

**Figure 19 marinedrugs-17-00350-f019:**
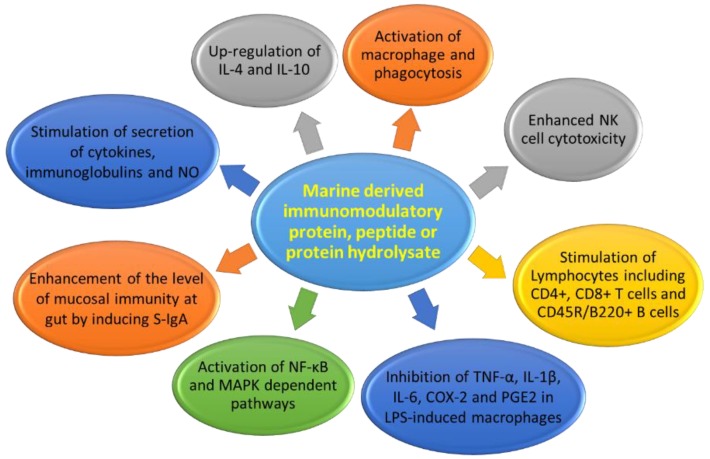
Mechanisms of the immunomodulatory effect from marine-derived protein, peptide, or protein hydrolysates.

**Table 1 marinedrugs-17-00350-t001:** List of immunomodulatory proteins from diverse marine sources.

Name of Protein	Source	Protein Type	Mechanism of Action	Ref.
Hemocyanin	Mollusk: *Concholepas concholepas, Megathura crenulata,* *Fissurella latimarginata*	Oxygen carrying metalloprotein	Immunostimulatory activities against certain cancers without side effects; interact with T cells, monocytes, macrophages, and polymorphonuclear lymphocytes to improve the host immune response	[23,24,25,26,27,28]
Lectin	*Clam: Crenomytilus grayanus, Mytilus trossulus, Fissurella latimarginata*	Glycoprotein	C-type lectins recognize carbohydrates during the immune response. Tachylectins recognize pathogen associated molecules via phagocytosis or the lectin pathway of the complement system. The C-type lectins play a key role in carbohydrate recognition during immune response. Lectins have been reported as pathogenic recognizing receptors from marine invertebrates. MTL stimulates the expression of proinflammatory cytokines (TNF-α and IFN-γ), but reduces the hyper-expressions of anti-inflammatory cytokine (IL-10).	[29,30,31,32,33,34]
Taurine	Clam: *Tapes philippinarum*	2-Amino ethane sulfonic acid	Cytoprotective and immunomodulatory effects in immune cells including lymphocytes, monocytes, and neutrophils; accumulation of phagocytes, contact with pathogens, activated cells (neutral and macrophages) produce toxic oxidants and various antibacterial substances using the peroxidase system and destroy the pathogens; scavenger to remove unwanted or harmful substances from the cells and protect them from oxidative stress; modulation of the immune system by activating NF-κB and activation PPAR-g.	[35,36,37,38,39,40,41,42,43,44,45,46,47,48,49,50,51,52,53,54,55,56,57,58,59,60,61,62,63,64,65,66,67,68,69,70,71,72,73,74,75,76,77]

**Table 2 marinedrugs-17-00350-t002:** Antimicrobial and immunomodulatory peptides from marine organisms.

Name of peptide	Source	Mechanism of action	Ref.		
Callinectin	Blue crab: *Callinectes sapidus*, Mediterranean mussel: *Mytilus galloprovincialis*	Antibacterial activity against gram-negative bacteria, binding to anti-callinectin-like peptides antibodies in blue crab hemocytes	[43,44]		
Clavanin A, clavanin-MO	Tunicate: *Styela clava*	Antimicrobial activity against Gram-negative, Gram-positive drug-resistant bacteria and fungi; immunomodulation by inhibiting the inflammatory response that causes sepsis and destroys certain biofilms; affect components of the immune system and influence inflammatory response; cytokine modulations (down-regulation of IL-12 and TNF-α, up-regulation of IL-1) in mice	[45,46]		
Crustin	Crustacean: *Carcinus maenas, Pacifastacus leniusculus, Fenneropenaeus chinensis, Scylla serrata, Scylla paramamosain, Penaeus monodon*	Antimicrobial activity against marine Gram-positive bacteria; release from the hemocytes of crustacean by exocytosis	[47,48,49,50,51]
Defensin	Oyster: *Crassostrea virginica*. *Crassostrea gigas* Mediterranean mussel: *Mytilus galloprovincialis* *M. edulis*	Antimicrobial peptides (AMPs) acting as host defense peptides that disrupt the membrane of microbial pathogens, and play a major role in immunomodulation by acting in the innate and adaptive immune response; after bacterial infection, increased MGD-1 in *M. galloprovincialis* plasma and MGD-2 stimulates release from hemocytes	[52,53,54,55]
Myticin	Mediterranean mussel: *Mytilus galloprovincialis*	Reached in the bacteria by transportation through hemocytes; antibacterial activity against Gram-positive bacteria (myticin A, myticin B, myticin C) and the fungus *Fusarium oxysporum* and *E. coli* (myticin C) and acts as immunomodulator in vivo; immune-related gene expression following in vivo immunostimulation in mussels	[56,57,58]
Mytilin	Mollusk: *Mytilus edulis* (mytilin A and mytilin B), *M. galloprovincialis* (mytilin C, mytilin D, mytilin G1)	Antimicrobial activities; transported through hemocytes to reach bacteria, and cells containing mytilin act as phagocytosing bacteria to prevent microbes from entering the circulatory system	[59,60]
Mytimycin	Blue mussel: *Mytilus edulis,* Mediterranean mussel: *M. galloprovincialis*	Antifungal activity; defense against invading pathogenic microbes; the gene responsible for mytomycin is mainly expressed in circulatory hemocytes	[55,59,61]
Phosvitin-derived peptide Pt5	Fish: *Danio rerio*	Antimicrobial activity and immunomodulatory activity; increase the survival rate of zebrafish infected by *Aeromonas hydrophila*, decrease the number of *A. hydrophila* in the blood, spleen, kidneys, liver, and muscles; inhibition expression of IL-1β, IL-6, TNF-α, and IFN-γ within the spleen and head kidneys of *A. hydrophila*-infected zebrafish, but increased the expression of IL-10 and IL-14	[62]
*Salmo salar* natural killer (NK)-lysin	Fish: *Salmo salar*	Antimicrobial activity; *Salmo salar* NK-lysin-derived peptides induce expression of IL-1β and IL-8 in *Salmo salar* head kidney leukocytes	[63,64]
Scygonadin	Mud crab: *Scylla serrata*	AMPs for host defense to protect the reproductive system of organisms	[65,66]
Thalassospiramides A and D	Bacteria: *Thalassospira* sp.	Suppression of LPS-induced NO production in RAW 264.7 macrophages; inhibition of IL-5 expression in TH-2-mediated inflammatory diseases such as asthma	[67,68]
Tilapia piscidin 3 (TP3) and tilapia piscidin 4 (TP4)	Fish: *Oreochromis niloticus*	Antimicrobial, wound-healing, and antitumor activity; increased expression of several immune-related genes in *O. niloticus* muscle (IL-1β, IL-6, IL-8, TGF-β, IκB), decreased expression of TLR5 after *Vibrio vulnificus* infection, down-regulation of IL-1β, IL-8 TLR5, TGF-β, and IκB after *Streptococcus agalactiae* infection	[69]

**Table 3 marinedrugs-17-00350-t003:** The amino acid sequences for marine-derived antimicrobial and immunomodulatory peptides.

Peptide	Source	Amino acid sequences	Ref./Genbank
Callinectin	*Callinectes sapidus*	WNSNRRFRVGRPPVVGRPGCVCFRAPCPCSNY-NH_2_	[43]
Clavanin-A, clavanin-MO	*Styela clava*	Clavanin A: VFQFLGKIIHHVGNFVHGFSHVF-NH_2_ Clavanin-MO: FLPIIVFQFLGKIIHHVGNFVHGFSHVF-NH_2_	[45,46]
Crustin	*Scylla serrata*	EASRVPPYLGRDCKHWCKDNNQALYCCGPPGITYPPFIRNHPGKCPSVRSTCTGVRSYRPKLCPHDGACDFRSKCCYDACVEHHVCKTV	[47] ADW11096.1
Defensin	*Crassostrea gigas*	GFGCPGNQSKCNNHCKSISCRAGYCDAATLWLRCTCTDCNGKK	[52] ACQ76262.1
Myticin C	*Mytilus galloprovincialis*	QSIPCTSYYCSKFCGSAGCSLYGCYKLHPGKICYCLHCRRAESPLALSGSARNVNEQNKE	[58] AEZ79080.1
Mytilin B	*Mytilus galloprovincialis*	SCASRCKGHCRARRCGYYVSVLYRGRCYCKCLRCSSEHSMKFPENEGSSPSDMMPQNENENTEFGQDMPTGETEQGETGI	[59] AAD45013.1
Mytomycin	*Mytilus edulis*	DCCHRPYYYHCWDCTAATPYCGYRPCNIFGCGCTCRTEPHGKSCYERGDRCRCYT	[61] AET85056.1
Phosvitin-derived peptide Pt5	*Danio rerio*	SRMSKTATIIEPFRKFHKDRYLAHHSATKDTSSGSAAASFEQMQKQNRFLGNDIP	[62]
*Salmo salar* NK-lysin	*Salmo salar*	KGTCWVCKWALNKVKKSISTSSSPETLKQKLLSVCDKVGFLKSMCKGLMKKHLWVLIEELSTSDDVRTICVNIKACKPKE	[63] XP_013985382
Scygonadin	*Scylla serrata*	GQALNKLMPKIVSAIIYMVGQPNAGVTFLGHQCLVESTRQPDGFYTAKMSCASWTHDNPIVGEGRSRVELEALKGSITNFVQTASNYKKFTIDEVEDWIASY	[65] AAW57403.1
Thalassospiramides A and D	*Thalassospira* sp.	cyclic lipopeptides contained rigid 12-membered ring containing an α,β-unsaturated carbonyl moiety	[67]
TP3 and TP4	*Oreochromis niloticus*	TP3: FIHHIIGGLFSVGKHIHSLIHGH, TP4: FIHHIIGGLFSAGKAIHRLIRRRRR	[69]

**Table 4 marinedrugs-17-00350-t004:** Immunomodulatory protein hydrolysates from diverse marine sources.

Name of hydrolysate	Source/amino acid sequence, MW	Treated enzymes	Mechanism of action	Ref.
*Chlorella* protein hydrolysate	Algae: *Chlorella vulgaris* (<5000 Da)	Pancreatin	Enhanced hemopoiesis, leukocyte count, peritoneal exudate cells, macrophage activity; stimulation of both humoral and cell-mediated immune functions (T-dependent antibody response and reconstitution of delayed-type hypersensitivity response) in BALB/c mice	[77]
*Ecklonia* protein hydrolysate	Algae: *Ecklonia cava*	Kojizyme^TM^	Increases in lymphocytes, monocytes, and granulocytes; increase in numbers of CD4+ T cells, CD8+ T cells, and CD45R/B220+ B cells; down-regulation of TNF-α and IFN-γ, up-regulation of IL-4 and IL-10 in ICR mice	[78]
*Porphyra* protein hydrolysate	Algae: *Porphyra columbina*	Alcalase®, trypsin, combination of both protease	Cytokine modulations (inhibition of TNF-α and IFN-γ, increase of IL-10) in rat splenocytes	[79]
*Porphyra columbina* protein hydrolysate	Algae: *Porphyra columbina*	Flavourzyme® and fungal protease concentrate	Immunomodulatory effects on rat macrophages and lymphocytes, activates NF-κB- and MAPK-dependent pathways, and mainly induces IL-10 production; inhibition of TNF-α, IL-1β, and IL-6	[80]
Edible red algae protein hydrolysate	Algae: *Porphyra tenera*	Alcalase®, Flavourzyme®, Neutrase®, Protamex^TM^, amyloglucosidase (AMG), Celluclast®, Dextrozyme®, Maltogenase, Promozyme, Termamyl®, Viscozyme®	Inhibition of LPS-induced NO production by murine macrophage RAW 264.7 cells	[81]
Edible microalgae Spirulina protein hydrolysate	Algae: *Spirulina maxima* LDAVNR (686 Da), MMLDF (655 Da)	Trypsin, pepsin, α-chymotrypsin	Inhibited histamine release and production from RBL-2H3 mast cells; interference with signaling pathways dependent on Ca^2+^ and microtubules (LDAVNR); inhibition of phospholipase Cγ activation and reactive oxygen species formation (MMLDF); NF-κB translocation and formation of IL-4	[82]
Oyster peptide-based enteral nutrition formula	Oyster: *Crassostrea hongkongensis*	Bromelain, pepsin, trypsin	Enhanced spleen lymphocyte proliferation and activity of NK cells in BALB/c mice	[83]
Oyster protein hydrolysate	Oyster: *Crassostrea gigas* (<3 kDa)	Protease from *Bacillus* sp. SM98011	Enhanced spleen lymphocyte proliferation; macrophage phagocytosis and NK cell cytotoxicity in BALB/c mice	[15]
*Paphia undulata* meat protein hydrolysate	Mollusk: *Paphia undulata* PHTC, VGYT, EF, LF and EGAL, WI, or WL	Protease from *Bacillus subtilis*	Enhanced mice spleen lymphocyte proliferation ability ex vivo	[84]
*Cyclina sinensis* protein hydrolysate	Venus clam: *Cyclina sinensis* RVAPEEHPVEGRYLV (1750.81 Da)	Pepsin	Enhanced macrophage phagocytosis, increased production of NO, TNF-α, IL-6, and IL-1β, and up-regulated protein levels of iNOS, NF-κB, and NLRP3 in RAW 264.7 cells; down-regulation of the expression of inhibitor of IκB-α; stimulation of macrophage activities by activating the NF-κB signaling pathway	[85]
*Rudi tapes* protein hydrolysate	Short-necked clam: *Ruditapes philippinarum* QCQQAVQSAV (876 Da)	Alcalase®, Flavourzyme®, Neutrase®, Protamex^TM^, α-chymotrypsin, papain, pepsin, trypsin	NO inhibitory activity in LPS-stimulated RAW 264.7 macrophages	[86]
Shellfish *Mytilus* protein hydrolysate	Shellfish: *Mytilus coruscus* GVSLLQQFFL (1151.37 Da)	Alcalase®, Flavourzyme®, Neutrase®, α-chymotrypsin, papain, pepsin, trypsin	Inhibited LPS-induced NO production in RAW264.7 macrophages	[87]
Alaska pollock protein hydrolysate	Alaska pollock: *Theragra chalcogramma* PYGADY (622 Da)	Trypsin	Enhanced humoral, cellular, and non-specific immunity in immunosuppressed mice	[88]
Alaska pollock frame protein hydrolysate	Alaska Pollock: *Theragra chalcogramma* NGMTY (584 MW), NGLAP (470 MW), and WY (305 MW)	Trypsin	Enhanced mice spleen lymphocyte proliferation activity	[89]
Fermented pacific whiting protein	Fish: *Merluccius merluccius* (<1 kDa)	Yeast	Enhanced phagocytic activity of peritoneal macrophages, increased number of IgA^+^ cells, and increased IL-4, IL-6, IL-10, IFN-γ, and TNF-α levels in the small intestine lamina propria in mice	[90]
Chum salmon oligopeptide preparation	Fish: *Oncorhynchus keta* (300–860 Da)	Complex protease	Enhanced lymphocyte proliferation capacity increased number of plaque-forming cells, increased NK cell activity, increased percentage of CD4^+^ TH cells in spleen and secretion of TH1 (IL-2, IFNγ) and TH2 (IL-5, IL-6)-type cell cytokines in ICR mice	[91]
Salmon fish protein hydrolysate	Fish: Atlantic salmon fish (Contained 60–70% di/tri peptides of < 10 kDa)	Endogenous hydrolyzing agents	Changes of IgM, IgG, and IgA and CD4/CD8 ratios were observed in malnourished Indian children	[92]
Salmon byproduct protein	Salmon fish byproduct from pectoral fin (1000–2000 Da)	Alcalase®, Flavourzyme®, Neutrase®, Protamex^TM^, pepsin, trypsin	Inhibited TNF-α, IL-6, and IL-1β in LPS-induced RAW264.7 macrophages	[93]
Salmon pectoral fin byproductprotein	Salmon fish byproduct from pectoral fin PAY (349.15 Da)	Pepsin	Inhibited production of NO and prostaglandin E2; production of pro-inflammatory cytokines, TNF-α, IL-6, and IL-1β in LPS-stimulated RAW264.7 cells	[94]
Shark-derived protein hydrolysate	PeptiBal^TM^, (innoVactiv, Inc.) (<10 kDa)	Trypsin, α-chymotrypsin	Enhanced gut barrier function via up-regulation of IgA-producing cells and intestinal cytokine production, including IL-6 and TNF-α in mice; inhibited production of TGF-β and IL-10 caused by infection with enterotoxigenic *E. coli* H10407	[95]
Sweetfish-derived protein hydrolysate	Sweetfish	Pepsin, trypsin, α-chymotrypsin	Inhibited production of NO, cytokines (TNF-α and IL-6), and PGE2 in LPS-induced RAW264.7 macrophages	[96]
Common carp egg protein hydrolysate	Fish: *Cyprinus carpio* egg (5-90 KDa)	Alcalase®, pepsin, trypsin	Enhanced proliferation of spleen lymphocytes, NK cell cytotoxicity, macrophage phagocytosis, level of mucosal immunity (S-IgA), and percentages of CD4^+^ and CD8^+^ cells in BALB/c mice	[97]
Rohu egg protein hydrolysate	Fish: *Labeo rohita* egg (<10 kDa)	Alcalase®, pepsin, trypsin	Significantly enhanced macrophage phagocytosis, NK cell cytotoxicity, mucosal immunity (S-IgA), splenic CD4^+^ & CD8^+^ T cells, and level of serum IgA in mice	[98]

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
