# Peer review of "Antimicrobial and Immunomodulatory Properties and Applications of Marine-Derived Proteins and Peptides"

_marinedrugs, 2019, doi:10.3390/md17060350_

Reviewer 1 Report

This is an exhaustive review on antimicrobial proteins and peptides from marine organisms. It is important to improve our knowledge on the potentiality of marine organisms as a source of AMPs.

Important to revise some points:

Check verb tenses: sometimes in a single paragraph there are simultaneously verbs in past and present tense.

Taurine (Table 1 and section 3.1.3) is not a protein, it is an amino acid. Cannot be included in the table and the protein section.

Primary sequences in table 2 should preferably be shown with a smaller letter size, and in a single lane.

Table 2. text corresponding to the column of mechanism of action should preferably be indicated schematically to spare space and facilitate the reading.

Overall, I think it is preferable to reduce the case size from sequences shown in figures 6,7,… so as to get the full sequence in a single line.

In the case of the peptides that have homologues in other non-marine organisms, it is preferable to include the information. For example, indicate the similarity between marine defensins and other vertebrate/invertebrate groups.

Likewise, highlight the proteins and peptides which are unique in marine organisms.

Overall, it will be important to include, when known, the information on potential toxicity of protein hydrolysates.

Lines 22-24: The abstract states that “most” peptides have undergone clinical trials?

There is not much information on clinical studies

Some sentences need rephrasing or corrections:

Line 22: “affected?”  the properties correlate with their amino acid composition

Line 23: mst immunomodulatory proteins and peptides?.  Better “proteins and peptides with immunomodulatory properties

Line 33: social pressure,… and pathogen presence. Better indicate first pathogen presence and then refer to other factors (better not to detail the potential social factors (not related to the present review).

Lin 44: include a reference(s) after penicllamine

Line 81: LPS? Heterosaccharides? Others specific for marine organisms?

Lin 88: first time to cite AMPs include full name.

Line 131: “However” is not correct here

Line 151: “previous” is not appropriate here

Line 152: “before the development?

Table 2:

Rephrase sentence: “Reached in the bacteria xxxx and act as phagocytosing bacteria”

Fish: zebrafish (better: zebrafish and latin name, to follow same criteria as other species in the table

Line 164: correct: cysteine residues, not cysteine molecules

Line 197: better indicate the disulphide bonds in the primary sequence in figure 6

Lines 217-218: a verb is missing

Line 240: cycle AMPs? However, figure 9 shows a linear peptide.

Line 253: indicate cys disulphide in the figure

Line 254: defends

Figure 10 indicates EF hand and Ca2+ motif and figure 12 indicates Saposin domain, but no information is provided in the text

Line 307: Thalassospiramides

Line 312: indicated by an elliptical circle… The other top left highlighted groups are not explained in the figure legend

Section on protein hydrolysates: better include in each section the origin and type of protease used: Kojizyme, flavourzyme,…

Line 345: compared to “in” non-supplemented

Line 364: inhibited (take out additional fullstop)

Check italics for all latin names.

Line 434: a parenthesis is missing

Lines 471-472: sentence is unclear: which are immunomodulatory? The amino acids or the peptides? Probably the peptides enriched with some amino acids?

Line 569: “can help reduce…” palliate?

Line 590: remove first line of references

Author Response

Response to Reviewer 1

This is an exhaustive review on antimicrobial proteins and peptides from marine organisms. It is important to improve our knowledge on the potentiality of marine organisms as a source of AMPs.

Important to revise some points:

Check verb tenses: sometimes in a single paragraph there are simultaneously verbs in past and present tense.

Authors’ Reply: Based on the comment from the reviewer, we checked verb tenses in the text.

Taurine (Table 1 and section 3.1.3) is not a protein, it is an amino acid. Cannot be included in the table and the protein section.

Authors’ Reply: We are agreeing with reviewer opinion. So, we included of immunomodulatory proteins and amino acids in the section 3.1 in the revised manuscript.

Primary sequences in table 2 should preferably be shown with a smaller letter size, and in a single lane.

Authors’ Reply: In Table 2, the column of amino acid sequence and mechanism of action are small, so we separated them into two tables (Table 2, mechanism of action and table 3, amino acid sequence) in the text.

Table 2. text corresponding to the column of mechanism of action should preferably be indicated schematically to spare space and facilitate the reading.

Authors’ Reply: In Table 2, the column of amino acid sequence and mechanism of action are small, so we separated them into two tables (Table 2, mechanism of action and table 3, amino acid sequence) in the text.

Overall, I think it is preferable to reduce the case size from sequences shown in figures 6,7,… so as to get the full sequence in a single line.

Authors’ Reply: Based on the comment from the reviewer, We redrawed the size of the sequence of Figures 6, 7, 8, 9, 10, 12, and 13 in the revised manuscript.

In the case of the peptides that have homologues in other non-marine organisms, it is preferable to include the information. For example, indicate the similarity between marine defensins and other vertebrate/invertebrate groups.

Likewise, highlight the proteins and peptides which are unique in marine organisms.

Authors’ Reply: In case of defensin, it was reported in animals and plants besides marine-organisms. Therefore, we have briefly mentioned these contents and the human-derived defensin in the revised manuscript.

(After modification)

Defensins are small cysteine-rich cationic AMPs that act as host defense peptides (Figure 7). Defensin was found in various sources, including animals, plants and insects. The human defensins are classified into the α-defensins and β-defensins on the basis of their sequence homology and their cystein residues. The first marine defensin was isolated by acidified gill extraction from Crassostrea virginica of American oyster [52, 53].

Overall, it will be important to include, when known, the information on potential toxicity of protein hydrolysates.

Authors’ Reply: Immunomodulatory protein hydrolysates have not been reported to be cytotoxic, unlike protein hydrolysates with antibacterial or anticancer activity. Based on the comment from the reviewer, we added to this sentence to the section 3.3 in the revised manuscript.

(After addition)

Immunomodulatory protein hydrolysates have not been reported to be cytotoxic, unlike protein hydrolysates with antibacterial or anticancer activity [70].

Lines 22-24: The abstract states that “most” peptides have undergone clinical trials?

There is not much information on clinical studies

Authors’ Reply: We are agreeing with reviewer opinion. Immunomodulatory proteins or peptides have been part of clinical trials, but this sentence has been confused. So, we've changed this sentence in the revised manuscript.

(After modification)

Proteins and peptides with immunomodulatory properties have been tested in vitro and in vivo, and some of them have undergone different clinical and preclinical pipeline.

Some sentences need rephrasing or corrections:

Line 22: “affected?”  the properties correlate with their amino acid composition

Authors’ Reply: Based on the comment from the reviewer, we modified this sentence in the revised manuscript.

(After modification)

The properties of immunomodulatory proteins and peptides correlate with their amino acid composition, sequence, and length.

Line 23: mst immunomodulatory proteins and peptides?.  Better “proteins and peptides with immunomodulatory properties

Authors’ Reply: As commented by the reviewer, we modified this sentence in the revised manuscript.

(After modification)

Proteins and peptides with immunomodulatory properties have been tested in vitro and in vivo, and some of them have undergone different clinical and preclinical pipeline.

Line 33: social pressure,… and pathogen presence. Better indicate first pathogen presence and then refer to other factors (better not to detail the potential social factors (not related to the present review).

Authors’ Reply: Based on the comment from the reviewer, we modified this sentence using the word ‘most of’ in the text. Following the reviewer's comments, we changed this sentence to another factor, leaving only pathogen presence among the factors that induce immune response in the revised manuscript.

(After modification)

The immune response plays a crucial role in maintaining human health by identifying and killing pathogens, aging cells, or tumor cells. Its functions can be affected by many factors, including pathogen presence, tissue injury, and cardiac infarction [1].

Lin 44: include a reference(s) after penicllamine

Authors’ Reply: Based on the comment from the reviewer, we added references (5-14) in the revised manuscript.

(Added references)

5. Gruppen, M.P.; Bouts, A.H.; Jansen-van der Weide, M.C.; Merkus, M.P.; Zurowska, A.; Maternik, M.; Massella, L.; Emma, F.; Niaudet, P.; Cornelissen, E.A.M.; Schurmans, T.; Raes, A.; van de Walle, J.; van Dyck, M.; Gulati, A.; Bagga, A.; Davin, J.C. A randomized clinical trial indicates that levamisole increases the time to relapse in children with steroid-sensitive idiopathic nephrotic syndrome. Kidney International 2018, 93, 510–518.

6. Ulrich, C.; Bichel, J.; Euvrard, S.; Guidi, B.; Proby, C.M.; van de Kerkhof, P.C.M.; Amerio, P.; Rønnevig, J.; Slade, H.B.; Stockfleth, E. Topical immunomodulation under systemic immunosuppression: results of a multicentre, randomized, placebo-controlled safety and efficacy study of imiquimod 5% cream for the treatment of actinic keratoses in kidney, heart, and liver transplant patients. Br. J. Dermatol. 2007, 157, 25-31.

7. Trabattoni, D.; Clerici, M,; Centanni, S.; Mantero, M.; Garziano, M.; Blasi, F. Immunomodulatory effects of pidotimod in adults with community-acquired pneumonia undergoing standard antibiotic therapy. Pulm. Pharmacol. Ther. 2017, 44, 24-29.

8. Ekins, S.; Lingerfelt, M.A.; Comer, J.E.; Freiberg, A.N.; Mirsalis, J.C.; O'Loughlin, K.; Harutyunyan, A.; McFarlane, C.; Green, C.E.; Madrid, P.B. Efficacy of tilorone dihydrochloride against ebola virus infection. Antimicrob. Agents Chemother. 2018, 62, e01711-01717.

9. Ahlmann, M.; Hempel, G. The effect of cyclophosphamide on the immune system: implications for clinical cancer therapy. Cancer Chemother. Pharmacol. 2016, 78, 661-671.

10. Cornejo-García, J.A.; Perkins, J.R.; Jurado-Escobar, R.; García-Martín, E.; Agúndez, J.A.; Viguera, E.; Pérez-Sánchez, N.; Blanca-López, N. Pharmacogenomics of prostaglandin and leukotriene receptors. Front. Pharmacol. 2016, 7, e316.

11. Flores, C.; Fouquet, G.; Moura, I.C.; Maciel, T.T.; Hermine, O. Lessons to learn from low-dose cyclosporin-A: a new approach for unexpected clinical applications. Front. Immunol. 2019, 10, e00588.

12. Jantan, I.; Ahmad, W.; Bukhari, S.N. Plant-derived immunomodulators: an insight on their preclinical evaluation and clinical trials. Front. Plant Sci. 2015, 6, e00655.

13. Ozkan, M.C.; Tombuloglu, M.; Sahin, F.; Saydam, G1. Evaluation of immunomodulatory drugs in multiple myeloma: single center experience. Am. J. Blood Res. 2015, 5, 95-100.

14. Gong, Y.; Frederiksen, S.L.; Gluud, C. D-penicillamine for primary biliary cirrhosis. Cochrane Database Syst Rev. 2004, 18, CD004789.

Line 81: LPS? Heterosaccharides? Others specific for marine organisms?

Authors’ Reply: According to the reviewer's comments, LPS was changed to polysaccharides because it was not appropriate in the revised manuscript.

(After modification)

Marine-derived substances, such as proteins, peptides, glycoproteins, polysaccharides, and lipids show potential for controlling the immune system [20-22].

Line 88: first time to cite AMPs include full name.

Authors’ Reply: Based on the comment from the reviewer, we added full name of AMPs in the revised manuscript.

Line 131: “However” is not correct here

Authors’ Reply: Based on the comment from the reviewer, we changed the word ‘However’ to ‘This suggests that’ in the text.

Line 151: “previous” is not appropriate here

Line 152: “before the development?

Authors’ Reply: we omit that this sentence does not match the previous sentence. Therefore, we have changed this sentence in the revised manuscript.

(After modification)

For this reason, it is necessary to understand the interaction between peptides and these systems for the development of specifc immunomodulatory peptides.

Table 2:

Rephrase sentence: “Reached in the bacteria xxxx and act as phagocytosing bacteria”

Authors’ Reply: Based on the comment from the reviewer, we removed the rephrase sentence (and act as phagocytosing bacteria) in the Table 2.

Fish: zebrafish (better: zebrafish and latin name, to follow same criteria as other species in the table

Authors’ Reply: Based on the comment from the reviewer, we changed the latin name of zebrafish to Danio rerio in the Table 2.

Line 164: correct: cysteine residues, not cysteine molecules

Authors’ Reply: Based on the comment from the reviewer, we changed ‘cysteine molecules’ to ‘cysteine residues’ in the revised manuscript.

Line 197: better indicate the disulphide bonds in the primary sequence in figure 6

Authors’ Reply: Based on the comment from the reviewer, we modified Figure 6 by adding six disulphide bonds in the primary sequence of crustin.

Lines 217-218: a verb is missing

Authors’ Reply: Based on the comment from the reviewer, we modified this sentence in the revised manuscript.

(After modification)

Following bacterial infection, an increase of MGD-1 in M. galloprovincialis plasma and MGD-2 stimulated the release from hemocytes.

Line 240: cycle AMPs? However, figure 9 shows a linear peptide.

Authors’ Reply: Mytilins are cysteine-rich cationic AMPs. So, we deleted the word ‘cyclic’ in the text.

Line 253: indicate cys disulphide in the figure

Authors’ Reply: Based on the comment from the reviewer, we modified Figure 10 by adding six disulphide bonds in the primary sequence of mytimycin.

Line 254: defends

Authors’ Reply: Based on the comment from the reviewer, we modified this word in the text.

Figure 10 indicates EF hand and Ca2+ motif and figure 12 indicates Saposin domain, but no information is provided in the text

Authors’ Reply: Based on the comment from the reviewer, we added the contents of EF hand and Ca2+ motif (mytimycin) and saposin domain (NK-lysin) to the text.

(After addition)

- Mytimycin, which consists of 12 cysteines connecting 6 disulfide bridges and C-terminal extension contains an EF-hand domain (Ca2+ binding motif), defends against invading pathogenic microbes.

- NK-lysin are AMPs is composed of 74–78 residues and contains six cysteine residues that form three disulfide bonds and C-terminal region contains an saposin B type domain (Figure 12).

Line 307: Thalassospiramides

Authors’ Reply: Based on the comment from the reviewer, we modified this word in the text.

Line 312: indicated by an elliptical circle… The other top left highlighted groups are not explained in the figure legend

Authors’ Reply: In Figure 14, the lipopeptide side chain includes both elliptical circles. Therefore, Figure regend was modified.

(After modification)

Figure 14. Chemical structures of thalassospiramide A (a) and thalassospiramide D (b). Lipopeptide side chain indicated two elliptical circles [57].

Section on protein hydrolysates: better include in each section the origin and type of protease used: Kojizyme, flavourzyme,…

Authors’ Reply: Based on the comment from the reviewer, we added the origin and type of protease used in the text.

Line 345: compared to “in” non-supplemented

Authors’ Reply: Based on the comment from the reviewer, we removed “in” in the text.

Line 364: inhibited (take out additional fullstop)

Authors’ Reply: Based on the comment from the reviewer, we changed in this sentence ‘inhibited’ to ‘decreased’ in the revised manuscript.

Check italics for all latin names.

Authors’ Reply: Based on the comment from the reviewer, we checked italics for all latin names in the text.

Line 434: a parenthesis is missing

Authors’ Reply: Based on the comment from the reviewer, we added a parenthesis in the text.

Lines 471-472: sentence is unclear: which are immunomodulatory? The amino acids or the peptides?

Probably the peptides enriched with some amino acids?

Authors’ Reply: We are agreeing with reviewer opinion. So, we've changed this sentence in the revised manuscript.

(After modification)

These protein hydrolysates show immunomodulatory activity with the highest relative lymphocyte proliferation activity. The main amino acid residues in the purified Alaska pollock frame protein hydrolysate were proline, aspartate, glutamic acid, and leucine, which the peptides enriched with main amino acid residues were confirmed to have immunomodulatory activity [89].

Line 569: “can help reduce…” palliate?

Authors’ Reply: Based on the comment from the reviewer, we changed in this sentence ‘can help reduce’ to ‘palliated’ in the revised manuscript.

Line 590: remove first line of references

Authors’ Reply: Based on the comment from the reviewer, we removed first line of references in the revised manuscript.

Reviewer 2 Report

Review of the article entitled: “Antimicrobial and immunomodulatory properties and applications of marine-derived proteins and peptides

Manuscript ID: marinedrugs-508059       

I have read the manuscript with a great interest. The manuscript is well written and organized. I have no doubt that the manuscript can be accepted for publication in “Marine drugs”. However, I have several (minor of importance) comments which should be taken into account before final acceptance. The detailed comments are presented below.

Abstract

I would propose to change the first sentence of the abstract.

Line 23 – in vitro and in vivo - should be written in italics

Introduction

Lines 46-47 – this conclusion is too general, not all naturally derived immunomodulatory proteins or peptides are absolutely safe. I would propose using a “key word” – “definitely most” of or “most of” - naturally derived immunomodulatory proteins or peptides.

Chapter 3

Line 85 – what do the authors mean by “Marine biomaterials” – it should be explained

Marine organisms are a rich source of taurin, but it should be clearly written that this substance is also produced by many other organisms, it is also synthesized – it also should be mentioned.

Line 198 – which marine Gram-positive bacteria do the authors mean – they should give some examples

Line 208 – a more general definition of defensins should be presented, these peptides are also produced by other organisms (not only marine organisms).

A general comment for a subchapter 3.3 - Immunomodulatory protein hydrolysates

I am a bit surprised (or even disappointed) that the authors did not take into account antimicrobial activity of hydrolysates produced from the proteins of marine organisms. I understand that the authors are not able to present all the health-promoting properties of protein hydrolysates. However, taking into account the title of the manuscript and the content of presented earlier chapters it should be at least mentioned – 3-4 sentences summarizing this issue would be required. Many researches in this area have been conducted and there are many publications which confirm antimicrobial properties of hydrolysates produced from proteins of marine organisms.

My final opinion is minor revision.

Author Response

Response to Reviewer 2

Review of the article entitled: “Antimicrobial and immunomodulatory properties and applications of marine-derived proteins and peptides

Manuscript ID: marinedrugs-508059       

I have read the manuscript with a great interest. The manuscript is well written and organized. I have no doubt that the manuscript can be accepted for publication in “Marine drugs”. However, I have several (minor of importance) comments which should be taken into account before final acceptance. The detailed comments are presented below.

Abstract

I would propose to change the first sentence of the abstract.

 Authors’ Reply: Based on the comment from the reviewer, this sentence is similar to the sentence in the latter part of abstract, so it was removed instead of changed in the revised manuscript.

Line 23 – in vitro and in vivo - should be written in italics.

 Authors’ Reply: We rewrote it in italics in the revised manuscript.

Introduction

Lines 46-47 – this conclusion is too general, not all naturally derived immunomodulatory proteins or peptides are absolutely safe. I would propose using a “key word” – “definitely most” of or “most of” - naturally derived immunomodulatory proteins or peptides.

Authors’ Reply: Based on the comment from the reviewer, we modified this sentence using the word ‘most of’ in the text.

(After modification)

Most of naturally derived immunomodulatory proteins or peptides do not show side effects and are less costly, suggesting their potential for use in immunotherapy.

Chapter 3

Line 85 – what do the authors mean by “Marine biomaterials” – it should be explained

Authors’ Reply: Based on the comment from the reviewer, the contents of ‘marine biomaterials’ were added to the beginning of section 3.1. in the revised manuscript.

(After addition)

3.1. Immunomodulatory proteins

Marine biomaterials (proteins, enzymes, oligosaccharides, biopolymers, fatty acids, minerals and pigments etc.) derived from marine natural source. Marine biomaterials contain large amounts of diverse proteins (10–47% (w/w)) with various bioactivities and functions.

Marine organisms are a rich source of taurin, but it should be clearly written that this substance is also produced by many other organisms, it is also synthesized – it also should be mentioned.

 Authors’ Reply: The sentence was added to the 3.1.3 session to refer to the fact that the source of taurine is produced from a variety of organisms according to the reviewer's comments.

(After modification)

3.1.3. Taurine

Taurine (2-amino ethane sulfonic acid) is an amino acid that is widely distributed in animal tissues including in marine clam (Figure 3). Marine organisms are a rich source of taurine, but it is also produced by many other organisms. Taurine has cytoprotective and immunomodulatory effects and is enriched in immune cells including lymphocytes, monocytes, and neutrophils [35].

Line 198 – which marine Gram-positive bacteria do the authors mean – they should give some examples

Authors’ Reply: Based on the comment from the reviewer, we added the name of the marine gram-positive bacteria in the revised manuscript.

(After modification)

 Crustins with specific activity against marine Gram-positive bacteria Corynebacterium glutamicum have been reported in various crustaceans such as C. maenas, Pacifastacus leniusculus, Scylla paramamosain, and Penaeus monodon (MW 7–14 kDa).

Line 208 – a more general definition of defensins should be presented, these peptides are also produced by other organisms (not only marine organisms).

Authors’ Reply: According to the reviewer's comments, we added that the defensein was found not only in marine organisms but also in various other organisms.

(After modification)

3.2.4. Defensin

Defensins are small cysteine-rich cationic AMPs that act as host defense peptides (Figure 7). Defensin was found in various sources, including animals, plants and insects. The first marine defensin was isolated by acidified gill extraction from Crassostrea virginica of American oyster [52, 53]. Defensins are antimicrobial peptides that disrupt the membrane of microbial pathogens and play a major role in immunomodulation by acting in the innate and adaptive immune response [54].

A general comment for a subchapter 3.3 - Immunomodulatory protein hydrolysates

I am a bit surprised (or even disappointed) that the authors did not take into account antimicrobial activity of hydrolysates produced from the proteins of marine organisms. I understand that the authors are not able to present all the health-promoting properties of protein hydrolysates. However, taking into account the title of the manuscript and the content of presented earlier chapters it should be at least mentioned – 3-4 sentences summarizing this issue would be required. Many researches in this area have been conducted and there are many publications which confirm antimicrobial properties of hydrolysates produced from proteins of marine organisms.

Authors’ Reply: We are agreeing with reviewer opinion. Since few protein hydrolysates have been reported until recently, both antibacterial and immunomodulatory activities, Section 3.3 summarizes the immunomodulatory protein hydrolysates. We also added a summary sentence for protein hydrolysates with antibacterial properties, taking into account the title of the manuscript and the content of presented earlier chapters.

(After modification)

3.3. Immunomodulatory protein hydrolysates

Protein hydrolysates derived from various proteins have been reported to have a wide range of biological activities, such as anti-inflammatory, anticancer, antioxidant, antimicrobial, anti-hypertensive, and immunomodulatory activities [17, 20, 70, 71]. During protein hydrolysis, peptide bond cleavage results in the formation of bioactive peptides with different sizes. Several proteolytic enzymes were successfully used to produce immunomodulatory protein hydrolysates. These enzymes include pancreatin, kojizyme, trypsin, alcalase, flavourzyme, protamex, αchymotrypsin, pepsin, neutrase, and thermolysin.

Immunomodulatory protein hydrolysates have not been reported to be cytotoxic, unlike protein hydrolysates with antibacterial or anticancer activity [70]. Antimicrobial peptides are important in the first line of the host defense system against pathogenic microorganisms that easily come in contact with the host through the environment [52, 53]. Antimicrobial peptides from marine protein hydrolysates are increasingly isolated and reported during the last few years [72-76]. Since there are few reports of marine derived protein hydrolysates having both antibacterial and immunomodulatory activity, this review summarizes the immunomodulatory protein hydrolysates isolated from diverse marine sources (Table 4).

(Inserted reference)

72. Ennaas, N.; Hammami, R.; Beaulieu, L.; Fliss, I. Purification and characterization of four antibacterial peptides from protamex hydrolysate of Atlantic mackerel (Scomber scombrus) by-products. Biochem. Biophys. Res. Commun. 2015, 462, 195-200.

73. Tang, W.; Zhang, H.; Wang, L.; Qian, H.; Qi, X. Targeted separation of antibacterial peptide from protein hydrolysate of anchovy cooking wastewater by equilibrium dialysis. Food chem. 2015, 168, 115-123.

74. Song, R.; Wei, R.-B.; Luo, H.Y.; Wang, D.F. Isolation and characterization of an antibacterial peptide fraction from the pepsin hydrolysate of half-fin anchovy (Setipinna taty). Molecules 2012, 17, 2980-2991.

75. Beaulieu, L.; Thibodeau, J.; Bonnet, C.; Bryl, P.; Carbonneau, M.E. Detection of antibacterial activity in an enzymatic hydrolysate fraction obtained from processing of Atlantic rock crab (Cancer irroratus) by-products. Pharma Nutrition 2013, 1, 149-157.

76. Balakrishnan, B.; Prasad, B.; Rai, A.K.; Velappan, S.P.; Subbanna, M.N.; Narayan, B. In vitro antioxidant and antibacterial properties of hydrolysed proteins of delimed tannery fleshings: comparison of acid hydrolysis and fermentation methods. Biodegrad. 2011, 22, 287–295.